# Shifts in isoform usage underlie transcriptional differences in regulatory T cells in type 1 diabetes

Jeremy R. B. Newman[1,2], S. Alice Long [ID] [3], Cate Speake [ID] [4], Carla J. Greenbaum [ID] [4], Karen Cerosaletti[3], Stephen S. Rich [ID] [5], Suna Onengut-Gumuscu [ID] [5], Lauren M. McIntyre [ID] [2,6], Jane H. Buckner [ID] [3] & Patrick Concannon[1,2 ✉]

Genome-wide association studies have identified numerous loci with allelic associations to Type 1 Diabetes (T1D) risk. Most disease-associated variants are enriched in regulatory sequences active in lymphoid cell types, suggesting that lymphocyte gene expression is altered in T1D. Here we assay gene expression between T1D cases and healthy controls in two autoimmunity-relevant lymphocyte cell types, memory CD4$^+$/CD25$^+$ regulatory T cells (Treg) and memory CD4$^+$/CD25$^-$ T cells, using a splicing event-based approach to characterize tissue-specific transcriptomes. Limited differences in isoform usage between T1D cases and controls are observed in memory CD4$^+$/CD25$^-$ T-cells. In Tregs, 402 genes demonstrate differences in isoform usage between cases and controls, particularly RNA recognition and splicing factor genes. Many of these genes are regulated by the variable inclusion of exons that can trigger nonsense mediated decay. Our results suggest that dysregulation of gene expression, through shifts in alternative splicing in Tregs, contributes to T1D pathophysiology.

[1] Department of Pathology, Immunology and Laboratory Medicine, College of Medicine, University of Florida, Gainesville, FL 32601, USA. [2] University of Florida Genetics Institute, University of Florida, Gainesville, FL 32601, USA. [3] Center for Translational Immunology, Benaroya Research Institute at Virginia Mason, Seattle, WA 98101, USA. [4] Center for Interventional Immunology, Benaroya Research Institute at Virginia Mason, Seattle, WA 98101, USA. [5] Center for Public Health Genomics, University of Virginia, Charlottesville, VA 22908, USA. [6] Department of Molecular Genetics and Microbiology, College of Medicine, University of Florida, Gainesville, FL 32601, USA. ✉email: patcon@ufl.edu

Type 1 diabetes (T1D) is an autoimmune disease arising from the T cell-mediated destruction of the insulinogenic pancreatic β cells, resulting in complete dependence on exogenous insulin to maintain glucose homeostasis[1,2]. A substantial genetic contribution to the disorder is well-established[3–5]. Up to half of the genetic risk for T1D is attributed to the human leukocyte antigen (HLA) gene cluster on chromosome 6[6–8]. There are also more than 90 non-HLA chromosomal regions for which significant evidence of association with T1D exists[9]. Fine mapping with the ImmunoChip of non-HLA regions associated with T1D combined with Bayesian inference has established a set of highly credible, putatively causative SNPs at many of these loci[10]. However, only a few of these credible causative variants are located in the coding regions of genes. Interrogation of 15 chromatin states across 127 tissues at the chromosomal positions of these credible SNPs revealed a strong enrichment for transcriptional enhancer sequences active in lymphocytes and other immune-relevant tissues[10], suggesting that changes in transcriptional regulation may be the mode of action for many of these T1D risk loci. These results are consistent with the hypothesis that most genetic variants that contribute towards disease risk are located in non-coding regions of the genome and modify gene regulation rather than impacting directly on protein function.

Dysregulation of transcription has been implicated in many human diseases[11–19] and can take the form of changes in overall transcriptional abundance of specific genes between affected and unaffected individuals or through alternative splicing that leads to alterations in transcript production, rather than gene usage. Alternative splicing of several genes in lymphocytes has been shown to be modified by T1D-associated risk variants located in or near those genes[20–24]. In *UBASH3A*, a rare alternate allele (G) at rs56058322 in intron 9 confers protection against T1D and favors the production of a truncated, intron-retaining isoform[25,26]. In *PTPN22*, a rare missense alternate allele (G) at rs56048322 in exon 18 is associated with T1D risk and results in the expression of two novel transcripts[27,28].

In this study we systematically evaluate transcript events in subsets of CD4+ T cells to determine whether lymphoid transcriptional dysregulation, in the form of alternative splicing, contributes towards T1D pathology. We broadly examine gene expression, splicing and isoform usage in subpopulations of memory CD4+ T cells, fractionated on their expression of CD25 (memory CD4+/CD25+ Tregs, memory CD4+/CD25− T cells) in order to elucidate transcriptional mechanisms underlying T1D in these cell types. Both cell types are relevant to T1D. Persistence of autoreactive memory T cells in T1D likely contributes to disease progression and limits the efficacy of immunomodulatory treatments or islet transplantation as therapies, while Tregs would normally be expected to control the autoimmune destruction of pancreatic beta cells that underlies T1D.

## Results

**Defining cell type-specific reduced reference transcriptomes.**
After excluding samples that failed quality control (low sequence coverage/quality, low RNA quality, ambiguous sample identity, etc.), sufficient quality RNA for sequencing and analysis was obtained from memory CD4+/CD25+ T cells (henceforth referred to as Tregs) for a total of 84 subjects (49 T1D cases and 35 controls), and from memory CD4+/CD25− T cells for 105 subjects (53 T1D cases and 52 controls). Characteristics of the subjects included in the analysis are summarized in Supplementary Table 1. An overview of the approach used to analyze these data are presented in Fig. 1. We utilized the method of Event Analysis[29] to construct reduced reference transcriptomes, which consist of transcripts that have all their exonic and junction sequences detected in either type 1 diabetic cases or unaffected controls in each cell type assayed. These approximate the set of expressed annotation-based transcripts for which there is evidence of expression. This is a data-driven approach that segments genes into their constitutive exonic sequences and exon-exon junctions; a transcript is excluded on the basis that there are one or more of their exons or junctions without sufficient sequencing coverage supporting their transcription.

The majority of detected transcriptional events were observed in both T1D cases and unaffected controls (Fig. 2a), and most events could be annotated to known transcripts (i.e., exon fragments and previously reported junctions), across the interrogated cell types. However, cell type specific differences were observed among unannotated events, i.e., previously undescribed junction and exon-intron border junctions. Unannotated events were more likely to be group specific in Tregs than in memory CD4+/CD25− T cells (Fig. 2a), and among Tregs there were more unannotated events detected in T1D cases than controls. This suggests that there is altered and potentially dysregulated transcription in Tregs from patients with T1D.

**Differences in expression between T1D case and control transcriptomes.** Almost all transcripts (Fig. 2b) and genes (Fig. 2c) in the reduced references were detected (TPM (transcripts per million) > 0) in both T1D cases and controls. Transcripts that were only detected in cases or only detected in controls were generally of low abundance (Supplementary Fig. 1). There were few genes represented in the reduced reference transcriptomes of each cell type that were significantly different between T1D cases and controls (FDR-corrected $P < 0.05$; Fig. 2c) in terms of total gene expression. More genes were differentially expressed in memory CD4+/CD25− T cells (195 of 7719 genes, 2.5%) than in Tregs (18 of 8443 genes, 0.2%; Fig. 2c), although generally there were few differences.

To provide validation for the cell population we defined here as Tregs and to determine if the gene expression differences observed, even if modest, were reflected in protein expression differences, we measured the mean fluorescence intensity (MFI) of several immune markers characteristic of Tregs on cells independently purified by flow cytometry (CD4+/CD25+/CD127lo) from the same peripheral blood mononuclear cell samples from which RNA for RNA-seq analysis was prepared (Supplementary Fig. 2). While some of these genes and their corresponding protein products did not differ significantly between T1D cases and controls, in all cases, the direction of the effect (i.e., higher expression in one population as compared to the other) was consistent for gene expression and MFI. One differentially expressed gene of note was *FOXP3*, a critical transcription factor in Tregs that is also one of their defining immunological markers: a modest increase in *FOXP3* gene expression was observed in T1D cases relative to controls (fold change = 1.42, $F = 7.46$, $P = 0.007$; Fig. 2d), and there was a correspondingly small but significant increase in FOXP3 protein expression in T1D cases (fold change = 1.07, $F = 7.92$, $P = 0.006$; Fig. 2e).

**Differential splicing in Tregs between T1D cases and controls.** Differential splicing was examined between T1D cases and controls, restricting our analysis to only multi-transcript genes defined as those with at least two transcripts in the reduced references. Thirty percent of the 8461 genes in the reduced reference in Tregs were multitranscript genes, as were 28% of 7733 genes in CD4+/CD25− T cells. In memory CD4+/CD25− T cells, only 0.3% of multi-transcript genes provided evidence of differential splicing. In contrast, 16% of the multi-transcript genes

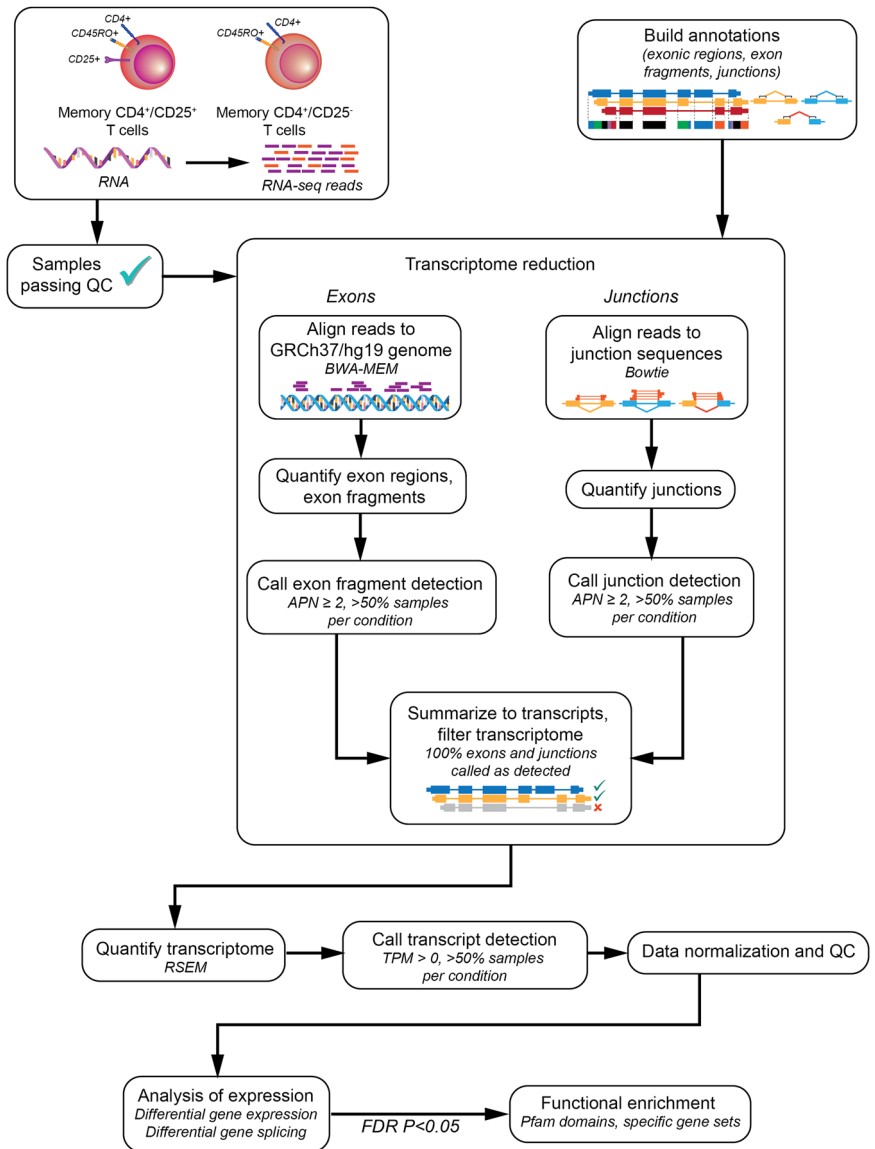

**Fig. 1 Overview of the analyses used in this study.** RNA is extracted from memory CD4$^+$/CD25$^+$ regulatory T cells and CD4$^+$/CD25$^-$ T cells of type 1 diabetic (T1D) patients and unaffected controls and sequenced. Samples passing QC are aligned to the GRCh37/hg19 human genome (for exonic sequences) and a database of all possible, logical junctions generated from the Aceview (2010 release) human genome annotations. Exons and junctions are quantified in each sample. For each condition (cell type × disease status), exonic sequences and junctions with an average depth per nucleotide (APN) of 2 or greater in at least 50% of samples are considered detected. Detected exons and junctions are summarized to transcripts and transcripts that do not have all their associated exons and junctions detected are filtered out, resulting a reduced set of transcripts per condition. Control- and T1D-specific reduced transcriptomes for each cell type are combined and quantified for each sample of that cell type. For each cell type, transcripts with a transcripts per million (TPM) estimate >0 in 50% of controls and/or T1D cases are considered detected and carried through to data normalization and additional sample QC. Following this, analysis of differential gene expression and differential gene splicing are carried out, and those genes considered statistically significantly different between controls and T1D cases are then further analyzed for functional domain enrichment, gene set enrichment, and additional analyses.

in Tregs were differentially spliced, with significant differences between T1D cases and controls (Fig. 2c). The most frequent splicing events in the differentially spliced genes in Tregs with the largest changes in percent-spliced-in (denoted as Ψ), the proportion that a particular splicing event is retained in the mature transcript, were those consistent with intron retention (Supplementary Data 1).

We next examined if there was evidence of differential exon fragment usage between T1D cases and controls. This is a variation on the test for differential splicing: while the gene-based differential-splicing test is based on the distribution of transcript levels within a gene and how this varies between T1D cases and controls, the test

for differential exon fragment usage considers if the distribution of exon levels within a gene vary. The test for differential exon fragment usage was examined to determine if (1) the differential exon fragment usage observed in the 403 differentially spliced genes in Tregs extended globally to all genes with exonic expression regardless of inclusion in the reduced reference transcriptomes; and (2) if there were more differential exon fragment usage between cases and controls observed in Tregs than in memory CD4$^+$/CD25$^-$ T cells. We observed an increased frequency of differential exon fragment usage (7.1%) in Tregs (1112 of 15,569 genes; Fig. 2f) compared to memory CD4$^+$/CD25$^-$ T cells (1.2%; 193 of 15,512 genes; Fig. 2f). Only 6% of genes exhibiting differential exon

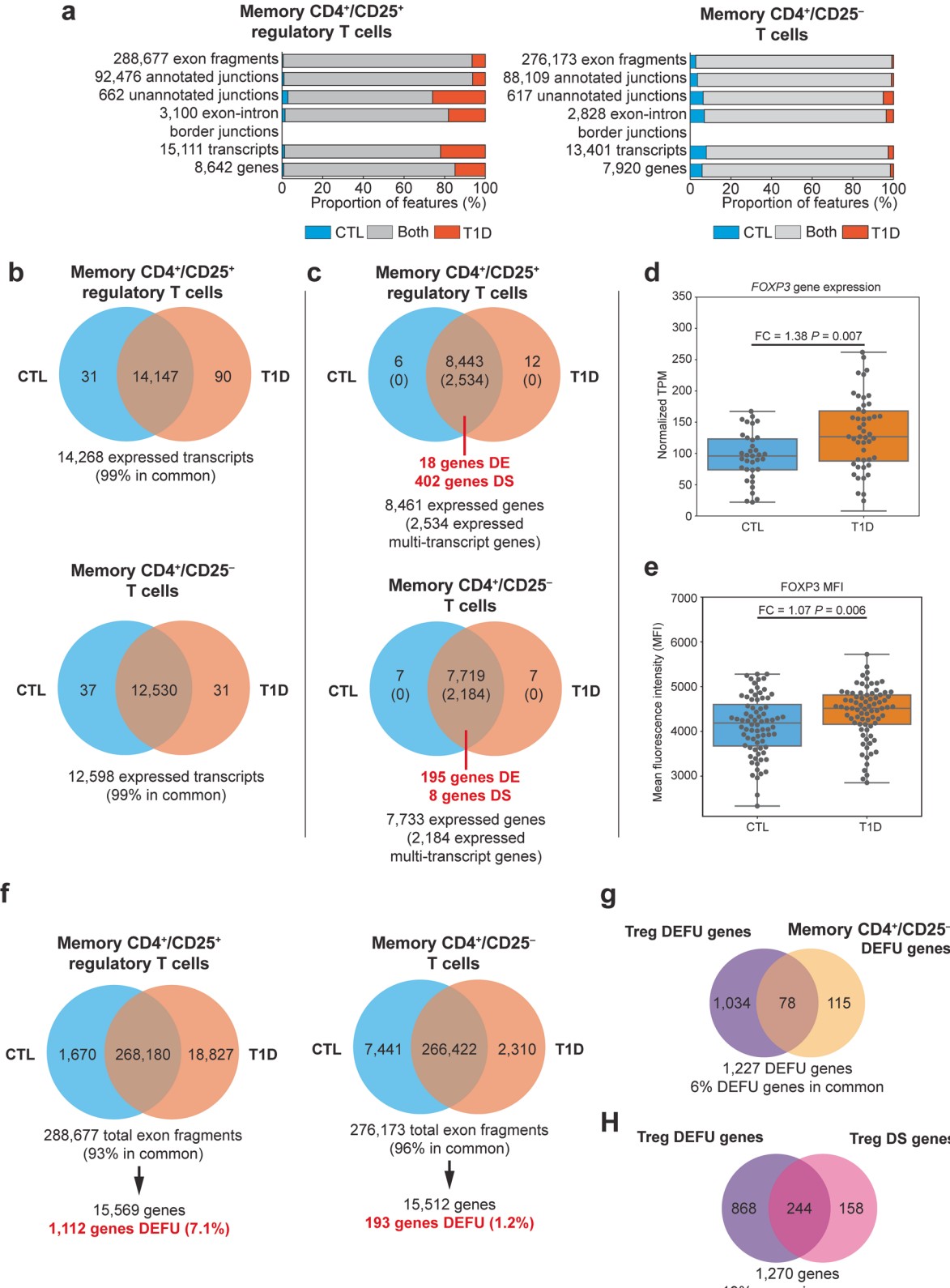

fragment usage were common to both cell types, consistent with the observed Treg-specificity of differential splicing (Fig. 2g). Compared to the set of differentially spliced genes in Tregs (Fig. 2c), 244 of the 402 (61%) genes with differential isoform usage in Tregs also had significant differential exon usage in Tregs (Fig. 2h), suggesting that changes in isoform structure are abundant in Tregs in T1D. Interrogation of unannotated transcriptional events (unannotated

junctions and exon-intron border sequences) revealed that in Tregs there was a higher fraction of genes with unannotated events exclusive to T1D cases, also indicative of altered splicing (Supplementary Fig. 3; Supplementary Note 1). These observations indicate that splicing is altered in Tregs in patients with T1D through shifts in transcript expression or through the inclusion/exclusion of specific exon sequences.

**Fig. 2 Summary of gene expression and splicing analysis. a** Transcriptional events—exon fragments, exon-exon junctions, and exon-intron border junctions—detected at average depth per nucleotide ≥2 in T1D cases (orange), unaffected controls (CTL; blue), and in both (gray) for memory CD4$^+$/CD25$^+$ Tregs and memory CD4$^+$/CD25$^-$ T cells, and the resulting reduced transcript sets with all associated events detected. Counts and summary data are available in Supplementary Data 2. **b** Transcripts detected at transcripts per million (TPM) > 0 in Tregs and memory CD4$^+$/CD25$^-$ T cells at TPM > 0. **c** Genes detected at TPM > 0. Numbers of detected multi-transcript genes are presented in parentheses. The number of significantly differentially expressed genes and differentially spliced genes (FDR P < 0.05) for each cell type is displayed in red text. DE = differentially expression, DS = differentially spliced **d** Distribution of normalized TPM for the *FOXP3* gene in memory CD4$^+$/CD25$^+$ Tregs (CTL: N = 35, median TPM = 96·13, interquartile range = 74.06–123.40; T1D: N = 48, median TPM = 126.95, interquartile range = 87.99–168.15). **e** Distribution of mean fluorescent intensity (MFI) of the FOXP3 protein in memory CD4$^+$/CD25$^+$ Tregs. FC = fold change calculated as the mean type 1 diabetics (T1D) value divided by the mean control (CTL) value (CTL: N = 75, median MFI = 4187.58, interquartile range = 3676.50–4601.99; T1D: N = 78, median MFI = 4517.04, interquartile range = 4158.49–4813.18). **f** Exon fragment detection and gene differential exon fragment usage test for Tregs and memory CD4$^+$/CD25$^-$ T cells. DEFU = differential exon fragment usage. **g** Comparison of genes with differential exon fragment usage between Tregs and memory CD4$^+$/CD25$^-$ T cells. **h** Comparison of genes with differential exon fragment usage (N = 1112) and quantitatively differentially spliced genes (N = 403) in Tregs. Data for boxplots are available in Supplementary Data 3 (**d**) and Supplementary Data 4 (**e**). Upper error bars are calculated as the third quartile + 1.5× interquartile range, lower error bars are calculated as first quartile − 1.5 × interquartile range.

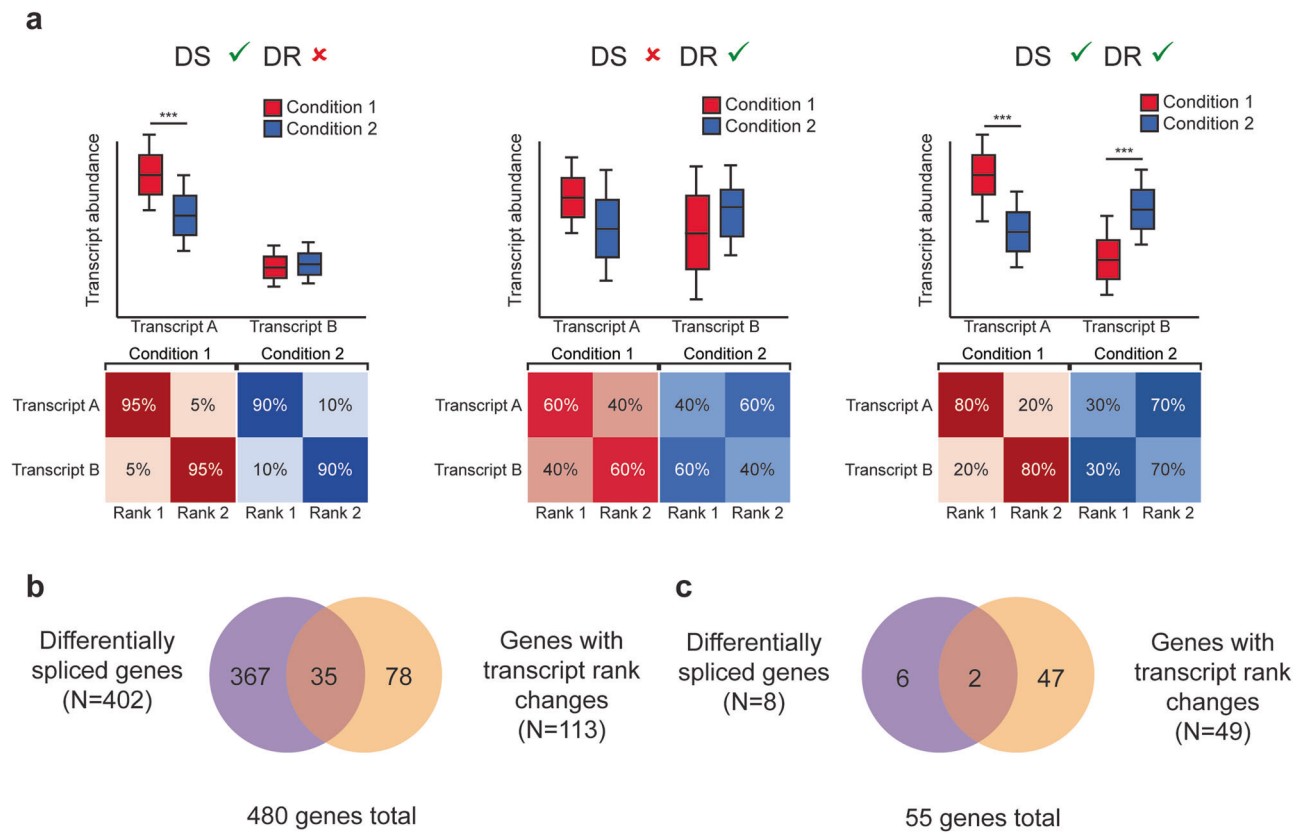

**Fig. 3 Results of test for differentially ranked transcripts. a** Hypothetical example of genes with quantitative changes in splicing (differentially spliced, DS) and/or changes in transcript ranking (differentially ranked, DR) between two conditions. Asterisks (***) indicate a hypothetical significant quantitative difference. Summary of transcript rank test and comparison with quantitative differential splicing test for multi-transcript genes in (**b**) Tregs (N = 2534 multi-transcript genes) and **c** memory CD4$^+$/CD25$^-$ T cells (N = 2184 multi-transcript genes).

**Differential splicing alters isoform usage**. In addition to determining whether isoforms are expressed at different levels, we also assessed whether the most common isoform changed between cases and controls by binning the transcripts of each gene into three categories (transcript ranks): rank 1 containing the most expressed transcript(s) of that gene, rank 3 comprising the least expressed transcripts(s), and rank 2 containing all other expressed transcripts. Figure 3a shows a hypothetical example of the difference between the differential splicing test and changes in transcript rank. A gene may have a statistically significant quantitative difference in transcript levels between T1D cases and controls, but transcripts may not necessarily deviate substantially in their transcript rank (Fig. 3a). Alternatively, there may a

scenario where the distribution of transcript levels of a gene do not differ significantly between conditions (e.g., similar mean expression and/or high variance) and therefore is not differentially spliced, but where there may be a significant change in how frequently transcripts are ranked (Fig. 3a). A gene may be considered differential spliced and also have differentially-ranked transcripts if there is a significant quantitative difference in transcript abundances and also a shift in transcript rank frequency (Fig. 3a).

Significant changes in transcript rank between T1D cases and controls were more abundant in Tregs than in memory CD4$^+$/CD25$^-$ T cells (FDR corrected P < 0.05; Fig. 2). In Tregs, 9% of differentially spliced genes also had significant changes in

transcript rank (Fig. 3b) suggesting that isoform switching is not a common cause of altered splicing in Tregs in T1D and is likely and primarily driven by changes in transcript abundances. A majority, 83 of the 113 (73%) genes with differentially-ranked transcripts had at least one transcript with a rank frequency difference ≥20% between T1D cases and controls, suggesting that the shift in isoform preference is large for those few genes in Tregs with significant changes in transcript rank.

In memory CD4$^+$/CD25$^-$ T cells there were 49 genes with differentially ranked transcripts, and 2 corresponded to genes that were differentially spliced between T1D cases and controls (Fig. 2b). Seven of the 49 (14%) genes with significant changes in transcript rank had at least one transcript with a large shift in isoform preference (rank frequency difference ≥20% between T1D cases and controls). This further demonstrates the Treg-specific bias of altered splicing in T1D and suggest that isoform switching may underlie some of these differences.

**Differentially spliced genes in Tregs are functionally enriched for RNA recognition motif.** We next investigated if certain sets of functionally-related or disease-related genes were over-represented among the genes differentially spliced in Tregs. No enrichment for genes located in chromosomal regions associated with T1D (12 of 95 (13%) T1D-associated region genes DS; 390 of 2439 (16%) non-T1D-associated region genes differentially spliced, $\chi^2 = 0.77$, DF = 1, $P = 0.38$) or with any autoimmune disorder (56 of 372 (15%) autoimmune disease-associated region genes differentially spliced; 346 of 2,162 (16%) non-autoimmune disease-associated region genes differentially spliced, $\chi^2 = 0.21$, DF = 1, $P = 0.64$) was observed.

We tested whether there were any functional protein family motifs that were more likely to be present in proteins encoded by differentially spliced genes in Tregs. Genes that encode at least one RNA recognition motif 1 (RRM-1; Pfam accession ID# PF00076.15) were significantly over-represented among differentially spliced multi-transcript genes in Tregs, regardless of the splicing test (24 of 67 (36%) RRM-1-containing multi-transcript genes differentially spliced, 378 of 2467 (15%) other multi-transcript genes differentially spliced, FDR-corrected $P = 0.023$). Genes containing RRM-1 domains were also more likely to demonstrate differential exon fragment usage than non-RRM-1 genes in Tregs (50 of 178 (28%) of RRM-1 genes with differential exon fragment usage; 1062 of 15,391 (7%) of all other genes with differential exon fragment usage; $\chi^2 = 99.94$, DF = 1, $P = 1.57 \times 10^{-23}$).

Because RNA recognition motifs are abundant among splicing factors[30], we tested whether splicing factor genes were enriched amongst differentially spliced genes and genes with differential exon fragment usage. Of the 67 known human splicing factor genes from SpliceAid[31] mapped to AceView identifiers, 47 were represented in the reduced reference transcriptome for Tregs, and 40 of these were genes expressing multiple transcripts (1.4% of 2534 multi-transcript genes). Many of these multi-transcript splicing factor genes exhibited differential splicing between T1D cases and controls in Tregs (18 of 40 (45%) differentially spliced splicing factor genes, 384 of 2494 (15%) of other multi-transcript genes differentially spliced, $\chi^2 = 25.85$, DF = 1, $P = 3.69 \times 10^{-7}$). No differentially spliced splicing factor genes were detected in memory CD4$^+$/CD25$^-$ T cells. When exons are tested directly without regard to transcript, splicing factor genes were over-represented in the genes with differential exon fragment usage (33 of 55 (60%) of splicing factor genes with differential exon fragment usage; 1079 of 15,514 (7%) of all other genes with differential exon fragment usage; $\chi^2 = 172.16$, DF = 1, $P = 2.49 \times 10^{-39}$). This feature was specific to Tregs, as no

RRM-1 domain-containing gene or splicing factor gene was considered to exhibit differential exon fragment usage in memory CD4$^+$/CD25$^-$ T cells. This suggests that splicing factors and other RNA recognition genes are alternatively spliced in Tregs from patients with T1D.

**Features of RRM-1 containing differentially spliced splicing factor genes in Tregs.** Nine of the 18 multi-transcript splicing factor genes differentially spliced in Tregs are regulated via the differential inclusion of a poison cassette exon, an exon containing a premature termination codon that is normally skipped, but when included in the mature mRNA can trigger nonsense-mediated decay[32–35]. These include four members of the serine/arginine-rich splicing factor family (*SRSF5, SRSF7, TRA2A, TRA2B*) and five heterogeneous nuclear ribonucleoproteins (*HNRNPA2B1, HNRNPD, HNRNPH1, HNRNPL* as Aceview gene *HNRNPLandECH1, HNRPDL*). In four cases, it was the differential usage of the poison cassette exon that caused these splicing genes to be classified as differentially spliced in our analyses — *HNRNPA2B1, SRSF5, SFSF7, TRA2A,* and *TRA2B (*the remaining five genes express poison cassette exon-containing transcripts). Thus, differential splicing of these genes can, by altering the availability of key splicing factors, dysregulate the splicing of a much broader set of genes, as we have observed in Tregs. Of note, ten of the dysregulated splicing factors are also either reported as FOXP3 target genes in Tregs (*HNRNPA2B1, HNRNPC, HNRNPD, HNRNPF, HNRNPK, SF1, SFSF5, TRA2A,* and *TRA2B*)[36] or FOXP3-interacting genes (*HNRNPL*)[37].

**Serine/Arginine-rich Splicing Factor 7 (*SRSF7*).** Among the RRM-1 domain-containing genes with perturbed splicing in T1D, we identified *SRSF7*, a gene that encodes a member of serine/arginine-rich splicing factor gene family and component of the spliceosome[38–41]. After excluding transcripts with incomplete exon or junction detection, only the *SRSF7.c* and *SRSF7.l* isoforms remained (Fig. 4a). Both isoforms are predicted to contain an RRM-1 domain in exon 2 and a C$_2$HC-type zinc knuckle domain in exon 3. The main difference between these two transcripts is the retention of intron 3 in *SRSF7.l*, which introduces a premature stop codon predicted to result in the production of a truncated SRSF7 protein. Part of intron-3 also encodes a well-studied poison cassette exon and the alternative splicing of intron 3 has been documented as an important regulator of *SRSF7* expression[38,41–43]. Indeed, both transcript and protein expression of a number of the SRSF family members, including SRSF7, have been shown to be modulated by differential poison cassette incorporation[43]. In Tregs from control subjects in our study, the intron-3-retaining *SRSF7.l* contributes little to the total *SRSF7* expression, while the *SRSF7.c* isoform is preferentially expressed and accounts, on average, for 75% of total *SRSF7* expression (Fig. 4b). In T1D cases, the expression of *SRSF7.l* increases and contributes ~40% of the total *SRSF7* expression (Figs. 4b, c), while the expression of the *SRSF7.c* isoform remains on average similar between cases and controls. Total *SRSF7* expression is only marginally higher in cases (Fig. 4c). The *SRSF7.l* isoform is frequently the most expressed transcript among T1D cases in Tregs, being preferentially expressed over *SRSF7.c* in 36% of cases compared to 14% in controls. Examination of genomic coverage demonstrates the increase in the retention of intron 3 (Fig. 4d, black arrow) without a corresponding decrease the intron-excluding *SRSF7.c* and largely coincides with the known *SRSF7* poison cassette exon (Fig. 4e). Expression of *SRSF7.c* and *SRSF7.l* was similar between T1D cases and controls in memory CD4$^+$/CD25$^-$ T cells (Fig. 4b, c), suggesting that T1D-associated changes in *SRSF7* regulation are specific to Tregs. Overall, this shows that

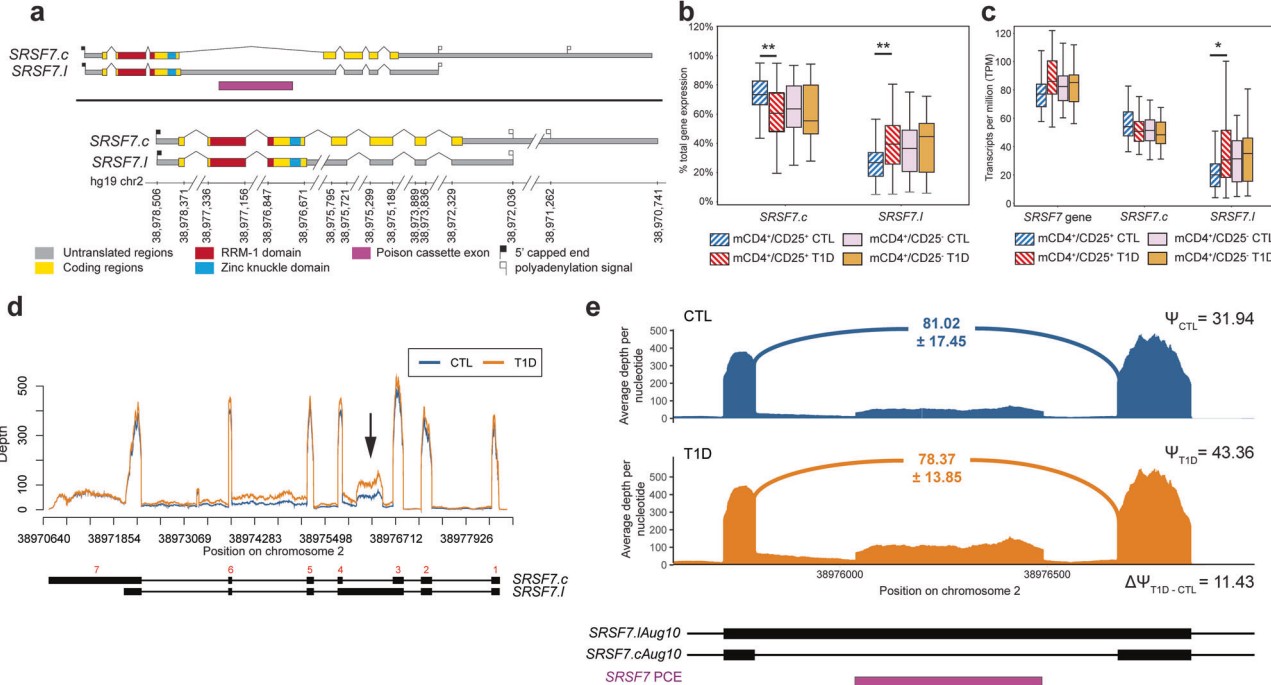

**Fig. 4 Differential splicing of *SRSF7* in Tregs from patients with T1D. a** *SRSF7* AceView transcripts represented in the EA-reduced transcriptomes for memory CD4+/CD25+ T cells and memory CD4+/CD25− T cells. Exonic sequences corresponding to Pfam domains are indicated. Intron and 3′UTR schematics have been reduced in size for visual clarity, as indicated by the broken line marks (/ /). **b** Distribution of the proportion of total *SRSF7* gene expression contributed by the *SRSF7.c* and *SRSF7.l* transcripts in Tregs between type 1 diabetes case (T1D) and unaffected controls (CTL) ($N = 35$ CTL, 48 T1D; *SRSF7.c* $F = 7.49$, $P = 0.008$, difference $= -10.95\%$; *SRSF7.l* $F = 7.49$, $P = 0.008$, difference $= 10.95\%$) and memory CD4+/CD25− T cells ($N = 50$ CTL, 57 T1D). **c** Distribution of transcripts per million (TPM) for total *SRSF7* gene expression and expression of *SRSF7.c* and *SRSF7.l* transcripts in Tregs ($N = 35$ CTL, 48 T1D; *SRSF7.l* $F = 1.71$, $P = 0.02$, fold change $= 1.71$) and memory CD4+/CD25− T cells ($N = 50$ CTL, 57 T1D). **d** Genomic coverage plot of *SRSF7* expression in Tregs. Red numbers above transcript models indicate exon numbering. Intron 3 is indicated by the black arrow. Red line is the average depth for controls; blue line is the average depth for cases; black transcripts are transcripts included in the Treg reduced reference transcriptome; grayed out transcript models are transcripts that were excluded from the reduced transcriptome reference for Tregs. **e** Detailed coverage plot of intron 3 retention showing the mean average depth per nucleotide of the intron 3-spanning junction for controls (blue) and cases (orange). The *SRSF7* poison cassette exon is annotated alongside *SRSF7* transcript models. Median and interquartile ranges for *SRSF7* expression and its transcripts (**c**, **d**) are available in Supplementary Data 5. Data for boxplots in (**c**) and (**d**) are available in Supplementary Data 6. *$P < 0.05$, **$P < 0.01$. Upper error bars are calculated as the third quartile + 1.5 × interquartile range, lower error bars are calculated as first quartile −1.5 × interquartile range.

there is an increase in the retention of intron 3 in Tregs from subjects with T1D and suggests the mechanism of *SRSF7* differential splicing in T1D Tregs is the retention of intronic sequences corresponding to a known poison cassette exon.

**Transformer 2 beta homolog (*TRA2B*).** Seven *TRA2B* transcripts annotated in AceView were included in the reduced transcriptomes for Tregs (Fig. 5a), although only two transcripts —*TRA2B.d* and *TRA2B.g*—contain an RRM-1 domain. *TRA2B* is a target of the transcription factor *FOXP3* which is critical for defining Tregs[36]. Transcripts *TRA2B.d*, *TRA2B.i*, *TRA2B.j* and *TRA2B.p* comprise the bulk of expressed *TRA2B* in Tregs (Fig. 5b, c). Small increases in the expression of *TRA2B.j* and *TRA2B.p* were seen in Tregs derived from T1D cases compared to controls (Figs. 5b and 5c). It has been reported that the second exon of *TRA2B*, which is present in transcripts "*TRA2B.j*" and "*TRA2B.p*", acts as a poison cassette exon[32,43] resulting in alterations in both transcript isoform distribution and protein expression. In Tregs, transcripts retaining this poison cassette exon are expressed at higher levels in T1D cases than in controls, while transcript "d" which skips the poison cassette exon is relatively unchanged (Fig. 5d, black arrow; Fig. 5e). As with *SRSF7*, no such differences in exon/isoform usage were observed in memory CD4+/CD25− T cells when comparing T1D cases and controls (Fig. 5b,c).

## Discussion

Aberrations in gene regulation via changes in alternative splicing and isoform usage have been implicated in the pathology of many complex disorders[11–19]. We have previously demonstrated, in a case-only study of T1D, that alternative splicing patterns displayed cell type specificity in lymphocytes, specifically in CD4+ T cells, CD8+ T cells and CD19+ B cells[26]. There are also several individual reports of changes in isoform production, in lymphocytes, regulated by genetic variants associated with T1D risk[20–28]. These findings suggest that changes in gene regulation, in the form of alternative splicing, may contribute toward the risk of T1D in a cell type-specific manner. As these effects can be detected in cells from individuals with long established disease it suggests that the alternative splicing we observe is not limited to the active phase of islet destruction in T1D.

Here we explore T1D case-control differences in splicing focusing on a subset of CD4+ T cells with particular relevance to autoimmunity. We compared two cell types with contrasting roles in autoimmunity, Tregs, which normally act to suppress autoimmunity by limiting the induction and proliferation of effector T cells[44] and memory CD4+ T cells that, if autoreactive, can sustain and promote autoimmunity[45]. Few differences were found between T1D cases and controls in memory CD4+/CD25− T cells, whereas several hundred genes exhibited evidence of differential splicing and isoform usage in Tregs. These alternative

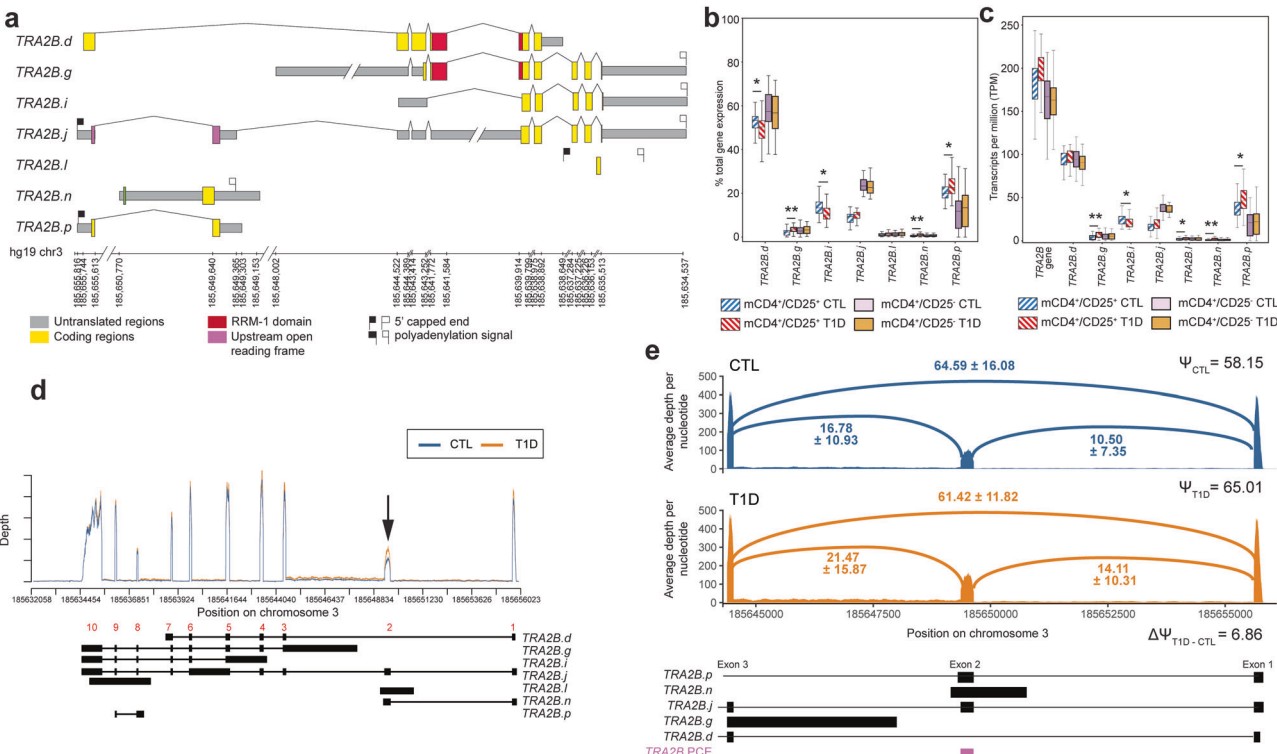

**Fig. 5 Differential splicing of *TRA2B* in Tregs from patients with T1D. a** *TRA2B* AceView transcripts represented in the Event Analysis-reduced transcriptomes for memory CD4+/CD25+ T cells and memory CD4+/CD25- T cells. Exonic sequences corresponding to Pfam domains are indicated. Intron and 3'UTR schematics have been reduced in size for visual clarity, as indicated by the broken line marks (/ /). **b** Distribution of the proportion of total *TRA2B* gene expression contributed by each expressed transcript in Tregs in type 1 diabetic cases (T1D) and unaffected controls (CTL) ($N = 35$ CTL, 48 T1D; *TRA2B.d* $F = 6.31$, $P = 0.01$, difference $= -3.48\%$; *TRA2B.g* $F = 8.18$, $P = 0.005$, difference $= 1.34\%$; *TRA2B.i* $F = 6.95$, $P = 0.01$, difference $= -2.52\%$; *TRA2B.n* $F = 8.33$, $P = 0.005$, difference $= 0.44\%$; *TRA2B.p* $F = 6.92$, $P = 0.01$, difference $= 3.00\%$) and memory CD4+/CD25- T cells ($N = 50$ CTL, 57 T1D). **c** Distribution of transcripts per million (TPM) for *TRA2B* gene and transcript expression in Tregs ($N = 35$ CTL, 48 T1D; *TRA2B.g* $F = 8.30$, $P = 0.005$, fold change $= 1.79$; *TRA2B.i* $F = 3.93$, $P = 0.05$, fold change $= -1.15$; *TRA2B.l* $F = 4.11$, $P = 0.05$, fold change $= 1.42$; *TRA2B.n* $F = 11.01$, $P = 0.001$, fold change $= 1.96$; *TRA2B.p* $F = 5.88$, $P = 0.02$, fold change $= 1.26$) and memory CD4+/CD25- T cells ($N = 50$ CTL, 57 T1D). **d** Genomic coverage plot of *TRA2B* expression in Tregs. Red numbers above transcript models indicate exon numbering. The poison cassette exon (exon 2) is indicated by the black arrow. Red line is the average depth for controls; blue line is the average depth for cases; black transcripts are transcripts included in the Treg reduced reference transcriptome. **e** Detailed coverage plot of exon skipping of *TRA2B* exon 2 showing the mean average depth per nucleotide of the exon-including and -excluding junctions for controls (blue) and cases (orange). *TRA2B* poison cassette exon is annotated alongside *TRA2B* transcript models. Median and interquartile ranges for *SRSF7* expression and its transcripts (**c, d**) are available in Supplementary Data 5. Data for boxplots in (**c, d**) are available in Supplementary Data 7. *$P < 0.05$; **$P < 0.01$. Upper error bars are calculated as the third quartile $+1.5 \times$ interquartile range, lower error bars are calculated as first quartile $-1.5 \times$ interquartile range.

splicing events have potentially major functional consequences and disease relevance as they include many examples of intron retention and are more frequently observed among T1D cases as compared to controls.

Most of the examples of intron retention we observed result in the introduction of an in-frame stop codon (79%) and/or a stop codon in all three reading frames[46]. The retention of these introns is likely to result in the production of truncated proteins with altered or absent function, and/or possibly target the transcript for nonsense mediated decay[33]. Total gene expression in Tregs is relatively consistent between T1D cases and controls; the primary difference between these subjects is in the ratio of isoform expression of these genes. Gene expression is largely unchanged, but the proportion that each transcript contributes to its gene's total expression is perturbed in T1D. The prevalence of intron retention and differential exon usage in differentially spliced genes in our study suggests that a critical mechanism may be the change in the ratio of functional (e.g., non-intron-retaining) transcripts to non-functional (e.g., intron-retaining) transcripts. We found that almost 50% of expressed splicing factor genes were differentially spliced in Tregs from patients with T1D; in many of

these genes, differential splicing can mechanistically be demonstrated by the inclusion of poison cassette exons and other regulatory sequences in mature transcripts in splicing factors and other RNA recognition genes, such as in *SRSF7*. Poison cassette exons are often ultra-conserved sequences, suggestive of their importance in gene regulation; in splicing-related genes the role of poison cassette exons is to autoregulate levels of functional protein isoforms and their impact upon both transcript and protein abundance is well documented[34,41–43,47,48]. In the *SRSF7* gene, intron 3 contains an in-frame stop codon that is more frequently retained in T1D cases than controls. This intron-retaining transcript contributes more towards the total *SRSF7* gene expression in T1D cases. Alterations in splicing were also identified by variation in transcript rank, resulting in different transcript isoforms being favored in T1D cases compared to controls.

Extending our investigation from transcripts to transcriptional events supported the hypothesis that dysregulated splicing in Tregs is related to the etiology of T1D. Changes in exon fragment usage/expression were observed between T1D cases and controls. We observed an increase in the expression of unannotated

transcriptional events likely to impact isoform structure, particularly unannotated junctions (aberrant exon skipping) and exon-intron border junctions (exon-intron read through). In Tregs, more of these events were detected in T1D cases than in controls, indicating the usage of alternative splice sites or simply the failure to efficiently recognize known splice sites. In contrast, there was little difference in the detection of unannotated events in memory CD4$^+$/CD25$^-$ T cells. The prevalence of intron retention and shifts in preferential isoform usage also suggest that in T1D the regulation of splicing is perturbed in Tregs (but not in memory CD4$^+$/CD25$^-$ T cells), perhaps through changes in isoform structure.

Several explanations are possible for these observations. Firstly, Tregs may be potentially in a state of cellular stress in T1D where changes in splicing are not a deliberate regulatory response to the disease state but instead incidentally manifest as malformed (or inefficient) splicing of several genes. Second, changes in splicing are in response to the disease environment of T1D in Tregs, whereby shifts in isoform usage serve to regulate gene expression at a post-transcriptional level by diverting proportions of transcripts for a given gene to isoforms that fail to produce fully functional protein. This mechanism could be active even where relatively few differentially expressed genes are observed (as is for Tregs). The abundance of intron retention, the presence of poison cassette exons among differentially spliced genes, and the increased expression of unannotated events in cases would seem to support this explanation. Alternatively, transcripts could produce truncated or altered proteins with the potential to disrupt cellular processes by dominant interference. Third, differential splicing in Tregs may reflect Treg phenotypic heterogeneity, or plasticity in their phenotype. Tregs can undergo a phenotypic reversion and a loss of their suppressive functions resulting in a phenotype similar to effector T cells[49–53]. These "ex-Tregs" have the ability to promote autoimmunity and expansion at sites of inflammation[49,54–56]. While this is associated with downregulation of FOXP3[54,57], differences in splicing observed in Tregs in T1D could reflect either a higher rate of phenotypic switching or a mechanism that could allow for more rapid switching between phenotypic states. However, the small (but statistically significant) increase in *FOXP3* gene and protein expression would suggest otherwise. Analysis of Tregs in T1D by single cell RNA-seq may resolve whether the differential splicing observed here is due to Treg heterogeneity in bulk RNA-sequencing data or is cell-intrinsic. Finally, as noted previously, nine of the splicing factor genes differentially spliced in Tregs have been shown to be targets of FOXP3. The elevated expression of *FOXP3* we observed in Tregs from T1D cases, as compared to controls, could contribute to the altered expression of downstream targets including these nine splicing factor genes, resulting in the elevated inclusion of poison cassette exons observed in some of these genes, such as *SRSF7*, without appearing to significantly alter the expression of protein-coding isoforms of these genes. This may reflect a regulatory switch that keeps the transcription of functional isoforms relatively consistent while shifting excess transcription to non-functional and/or poison cassette exon-containing transcripts that are degraded by nonsense mediated decay.

It is possible that the underlying mechanism in T1D may be a combination of these possible alternatives - altered splicing of some critical genes regulating the amount of functional transcript expressed as a response to stress or a consequence of phenotype switching. differentially spliced genes in Tregs are enriched for genes that encode RNA-binding proteins and known splicing factors. While splicing factor genes represent only a small fraction of all genes in the genome, ~45% of expressed splicing factor genes have significant changes in isoform usage in Tregs. This

suggests that protein-coding genes whose products regulate splicing may be themselves dysregulated in Tregs and consequently their dysregulation propagates and amplifies aberrant splicing patterns transcriptome-wide. This could lead to an autocatalytic response where the dysregulation of splicing-related genes leads to aberrant splicing of other splicing genes. The abundance of splicing factors that are differentially spliced in Tregs and that in several instances the splicing difference in question coincides with a reported poison cassette exon would suggest that this is a likely explanation for at least some of our observations. Such *trans* regulatory effects, driven by alternative splicing of splicing factor genes and/or other RNA-binding genes may be difficult to specifically delineate given the stochastic nature of transcription and the likely small effect sizes. Most of the shifts in isoform preference in our data are generally small (<20% difference in rank frequency) and are not readily distinguishable from possible *trans* effects. As regulation of alternative splicing may be crucial to proper Treg function[58,59], our findings suggest that the preferential expression of alternative transcripts and dysregulated splicing could alter Treg phenotype and function and, ultimately, contribute to T1D risk.

In summary, our findings highlight how alternative splicing and isoform usage can differentiate between T1D-relevant cell types as well as between subjects with and without T1D, even in the absence of significant overall gene expression differences. Our multiple approaches to examining splicing revealed changes in alternative splicing in T1D in Tregs, a critical cell type for maintaining peripheral immune tolerance. Many of the observed differences in isoform usage between T1D cases and controls in Tregs are in the form of changes in isoform structure and frequently involve the mis-splicing of transcripts encoding splicing factors, suggesting possible regulation through an autocatalytic mechanism acting on splicing to alter transcript abundance.

## Methods

**Subject ascertainment.** Subjects were ascertained from a study population of 77 T1D cases and 81 age- and sex-matched (non-T1D) controls. The mean age of all participants was 32.6 years (range: 18–49 years), with mean age of T1D onset in cases 19.2 years, with mean duration of disease 13.7 years (Supplementary Table 1). All samples were collected under protocols approved by the Benaroya Research Institute IRB (IRB-07109), with written informed consent obtained from all study participants. Methods were performed in accordance the relevant guidelines and regulations.

**Sample preparation and RNA sequencing.** CD4+CD25+ cell selections were performed using Miltenyi magnetic beads. Negative selection of memory CD4 T cells was performed on PBMC that were previously depleted of CD19$^+$ B cells and CD8$^+$ T cells. Memory CD4$^+$ T cells were collected using the Memory CD4$^+$ T cell Isolation Kit (Miltenyi) and fractionated into CD25$^+$ or CD25$^-$ subsets by positive selection using on CD25 Microbeads (Miltenyi). Sample purities were assessed approximately weekly during the collection period by flow cytometry (Supplementary Fig. 4).

Sufficient RNA for sequencing was purified from 44 controls and 55 T1D cases for memory CD4$^+$/CD25$^+$ T cells, and 66 controls and 67 T1D cases for memory CD4$^+$/CD25$^-$ T cells (Supplementary Table 1). RNA-seq libraries were prepared according to Illumina protocols and sequenced (2 × 101 nt reads) on an Illumina HiSeq 2000 instrument (137 million ± 58 million reads/sample). Quality of sequencing data was assessed using GC content, and the percentage of adapter content, duplication rate,

and homopolymer content in each sample (https://www.bioinformatics.babraham.ac.uk/projects/fastqc/)[60].

To confirm the general similarity of gene expression of samples of the same cell type, normalized expression counts for exonic regions (see Methods, Quantification of gene expression) were analyzed using hierarchical clustering and principal components analysis (JMP Genomics 7, SAS Institute). Expression data were centered and scaled by exonic region to mean of 0 and variance of 1. All parameters were left at their default settings. Samples that did not cluster with their cell type group were either samples of low coverage or otherwise flagged for removal.

To confirm subject sex and donor identity, variant calling was performed from the RNA sequencing data using the Genome Analysis Toolkit version 3.8.0[61] following the "best practices" for RNAseq short variant discovery. Reads were aligned to the human GRCh37 genome, and duplicate read sequences were marked using Picard 'MarkDuplicates' (https://broadinstitute.github.io/picard/). Base quality score recalibration was performed using dbSNP release 138, 1000 Genomes Phase 1 SNPs and indels, HapMap release 3.3 SNPs, and Mills and 1000 Genomes gold standard indels as reference variant sites. Genotypes were called running "HaplotypeCaller" in GVCF mode[61] and then using "GenotypeGVCFs" as recommended. Variant quality recalibration with a tranche threshold of 99.0% was then applied to minimize false positive calls; as genotype calls were used solely for the purpose of sample identity confirmation, novel variant discovery was not prioritized. Kinship coefficients were estimated using the KING algorithm implemented in PLINK[62,63] and used to assess genotype concordance between samples from the same individual (e.g., memory CD4$^+$/CD25$^+$ vs memory CD4$^+$/CD25$^-$) to confirm their common donor subject. Subjects were also previously genotyping with the ImmunoChip custom genotype array (Illumina) and the Axiom Precision Medicine Research Array (Thermo Fisher) as part of a T1D fine-mapping project[9]. Kinship coefficients were estimated between RNA sequencing samples and ImmunoChip samples to assess genotype concordance and confirm sample donor identity. Chromosome X and Y SNP genotype calls were used to confirm subject sex. In addition, the expression of the genes TISX and XIST (chromosome X genes involved in X-inactivation) and EIF1AY (chromosome Y) was also examined. The ratio of EIF1AY to TISX/XIST expression was calculated, where a high EIF1AY:XIST ratio indicated a male subject and a low or zero ratio indicated a female subject.

In total, 15 memory CD4$^+$/CD25$^+$ T cell samples (6 T1D cases, 9 controls) and 28 CD4$^+$/CD25$^-$ T cell samples (14 T1D cases, 14 controls) were subsequently excluded from further analyzes.

**Annotating transcriptional events**. We utilized the method of Event Analysis[29] to annotate the human transcriptome in terms of exons, exonic regions, individual sequence fragments within these exonic regions based on transcript membership (exon fragments), annotated exon-exon junctions, and all other possible, logical exon-exon junctions within a gene. We used AceView gene models for the hg19/GRCh37 genome to assess changes in expression and splicing due to the higher accuracy of its gene models over RefSeq and Ensembl[64].

**Quantification of gene expression**. Most transcripts in an annotation are unlikely to be expressed; genes not likely expressed in either memory CD4$^+$/CD25$^+$ Tregs or memory CD4$^+$/CD25$^-$ T cells were filtered[29] as were transcripts not detected in any condition. The remaining transcripts were used to quantify gene and transcript expression. A gene transfer format file was

generated consisting of only the exons of isoforms included in the reduced reference transcriptome. The program "gff_make_annotation" (https://github.com/yarden/rnaseqlib;[65]) was used to generate a set of annotations for alternative 3′ start sites, alternative 5′ start sites, mutually exclusive exons, skipped exons, and retained introns. Exon fragments and annotated junctions were assigned to each of these annotations as either inclusion events (supporting the inclusion of an alternative exon or exonic sequence from a transcript or transcripts) or exclusion events (supporting the exclusion of an alternative exon or exonic sequence from a transcript or transcripts). Individual transcriptional events (exonic regions, exon fragments, exon-exon junction, exon-intron border junctions) were quantified[29]. Sequencing reads were aligned to the generated database of junction reference sequences using Bowtie (version 0.12.9;[66]) to quantify junction coverage. Reads were aligned to the complete human genome (GRCh37/hg19 version) using the Burrows-Wheeler Aligner for short reads (BWA-MEM, version 0.7.12[67]) for coverage of exonic features, namely exonic regions, exon fragments, and introns[29]. Coverage was calculated for each event as the average depth per nucleotide. An event was considered detected if the average depth per nucleotide was ≥2 for more than half of all samples per group (cell type × case/control status); i.e., an average of 2 or more mapped reads per feature.

For the analysis of gene expression using transcripts, estimates of transcript abundance were obtained using RNA-seq by Expression-Maximization (RSEM version 1.2.28;[68]). Cell type-specific reduced transcriptome references were compiled by selecting transcripts with all constituent exons and junctions detected at an average depth per nucleotide ≥2 in either cases or controls[29]. Transcriptome references were prepared with 'rsem-prepare-reference' using the set of transcript sequences as FASTA sequence and a tab-delimited gene-to-transcript file as input[68]. Default settings were used for all parameters. Transcripts per million was the metric used to estimate transcript abundance due to its greater comparability between samples[69]. A transcript was considered expressed if the transcripts per million was >0 for >50% of all samples per group (cell type × case/control status).

**Differential gene expression and splicing**. To assess differential gene expression and splicing in genes represented in the reduced reference transcriptomes, for each gene, the data were modeled as Eq. 1:

$$Y_{ijklm} = \mu + t_i + d_j + (td)_{ij} + sk + (ds)_{jk} + p_l + V_m + \varepsilon ijklm \tag{1}$$

where $Y$ is the log-transformed normalized transcripts per million; $t$ is transcript $i$; $d$ is disease status ($j$ = control, case); $td$ is the interaction between transcript and disease status; $s$ is subject sex ($k$ = male, female); ds is the interaction between sex and disease status; $p$ is RNA-seq pool ($l$ = 1, 2, 3, 4, 5, 6; $i.i.d. \sim N(0, \sigma^2_P)$); $V$ is a matrix of latent factors for sample $m$ used to explain hidden confounders estimated using PEER factors;[70] and $\varepsilon$ is the residual ($\sim N(0, \sigma^2_P)$). For genes with only a single transcript, Eq. (1) reduces to Eq. (2):

$$Y_{jklm} = \mu + d_j + s_k + (ds)_{jk} + p_l + V_m + \varepsilon jklm \tag{2}$$

Only differential gene expression is tested for these single-transcript genes. RNA-seq pool was fit as a random effect to account for the within-pool variance, while all other factors were fit as fixed effects. This model was also used to assess differential expression of individual transcripts from all genes represented in the reduced reference transcriptomes.

If cases preferentially express different transcripts from controls, the F-test for the interaction between transcript and

disease status (*td*) will be significant, and the gene is considered differentially spliced[71]. If there is a difference in the overall expression of a gene, then the *F*-test for disease status (*d*) will be significant, and the gene is considered differentially expressed. Genes that are considered either differentially spliced or differentially expressed represent the main two hypotheses of interest[26,71,72]. *P* values were corrected for multiple tests using false discovery rate (FDR);[73] a FDR corrected $P < 0.05$ was considered as statistically significant. The model used to assess differential expression and differential splicing was applied to genes represented by detected exon fragments, with *Y* representing the log-transformed APN; *f* is exon fragment *i*; and *fd* is the interaction between exon fragment and disease status. All analyses of differential expression and differential splicing were conducted in SAS (v9.4; SAS Institute).

**Immune marker measurements**. MFI was measured for selected immune markers. Antibodies used are listed in Supplementary Table 2. All populations were gated for non-debris, live, singlet lymphocyte populations and then further for $CD4^+CD25^+CD127^{lo}$ or $CD4^+CD25^+FOXP3^+$ Treg, depending on the panel. MFIs were exported from cells in the Treg gate. Data were collected on a BD Fortessa cytometer using Diva software and analyzed with FlowJo software (version 7.2; TreeStar, Ashland, Oregon). Invitrogen 8 peak beads were used to normalize flow cytometry settings between experiments by adjusting voltage settings to reach a standard MFI.

To test whether levels of measured immunological markers were different between T1D cases and controls, MFI measurements were modeled was Eq. (3):

$$Y_{jklm} = \mu + d_j + s_k + (ds)_{jk} + b_l + \varepsilon jklm \quad (3)$$

where *Y* is the MFI; *d* is disease status (*j* = control, case); *s* is subject sex (*k* = male, female); *ds* is the interaction between sex and disease status; *b* is sample batch (*l* = 1, 2, 3, …, 10); and *ε* is the residual ($\sim N(0, \sigma^2_P)$).

**Identifying splicing differences between transcripts**. For each exon with annotated alternative splice variation and for each sample, we calculated the mean average depth per nucleotide of all inclusion events and the mean average depth per nucleotide of all exclusion events. From these, a percent spliced in score (*Ψ*)[65], was estimated as Eq. (4):

$$\Psi_{ij} = I_{ij}/(I_{ij} + E_{ij}) \quad (4)$$

where $I_{ij}$ is the mean average depth per nucleotide of all inclusion events for annotation *i* and subject *j* and $E_{ij}$ is the mean average depth per nucleotide of all exclusion events for annotation *i* and subject *j*.

A set of annotations was generated to examine the relative expression of alternative first and last exonic regions. The 5′-most (relative to strand) of the first/last exonic regions was considered as the reference exonic regions; all other first/last exonic regions were classified as alternative exonic regions. For each gene, all possible first and last exonic regions were derived from the set of the transcripts included in the reduced reference. We excluded any first or last exonic region that was also annotated to an internal exon of another transcript, due to ambiguity in defining reference exonic regions and alternative exonic regions. For each annotation and for each sample, we estimated Ψ of the alternative first/last exons using Eq. (5):

$$\Psi_{ij} = A_{ij}/(A_{ij} + R_{ij}) \quad (5)$$

where $A_{ij}$ is the mean normalized average depth per nucleotide of the alternative exonic region for annotation *i* and subject *j* and $R_{ij}$

is the mean normalized average depth per nucleotide of reference exonic region for annotation *i* and subject *j*. Splicing differences were annotated in terms of the exonic sequence being included/excluded. For mutually exclusive exons and alternative first and last exons, splicing differences were annotated as the 5′-most exon first, followed by the alternative exon. Statistically significant differences in Ψ of individual splicing events between T1D cases and controls were evaluated using a two-sided *t* test with unequal variances assumed.

**Transcript ranking**. A three-tiered ranking system was used to assess changes in isoform preference for genes with two or more transcripts by evaluating whether or not the transcript (or transcripts) that is (are) most/least expressed is (are) changing, without being confounded by small differences in abundance: i.e., sets of transcripts with similar estimates are grouped together. Within each sample and for each gene, transcripts were assigned to a rank of 1 if their estimated abundance was the highest or within 1 standard deviation of the highest (i.e., the set of most expressed transcripts); transcripts with a rank of 3 were those with the lowest estimated abundance, or within 1 standard deviation of the lowest (i.e., the set of least expressed transcripts); all other transcripts were assigned a rank of 2. For each group (cell type × case/control status), the frequency of each rank assignment for each transcript was calculated (rank frequency).

**Annotation enrichment**. Predicted protein family domains from the Pfam database[74] for human AceView transcripts were downloaded from the AceView website (ftp://ftp.ncbi.nih.gov/repository/acedb/ncbi_37_Aug10.human.genes/AceView.ncbi_37.pfamhits.txt.gz). Pfam domains were annotated to genes if they were present at least once in the coding region of that gene. Gene set enrichments were performed for all Pfam domains represented in the reduced reference transcriptomes using Fisher's exact test (JMP Genomics 9, SAS Institute), to identify the function of genes that were significantly more/less likely to be differentially expressed or differentially spliced compared to other expressed genes. Differences with an FDR-corrected $P < 0.05$ were considered statistically significant.

Enrichment tests targeted specific sets of genes, including RNA Recognition Motif 1 (RRM-1) domain-containing genes, splicing factor genes, autoimmune genes, FoxP3-target genes and FoxP3-interacting genes. Genes containing at least one RRM-1 domain were derived from AceView Pfam annotations using a list of human splicing factor genes obtained from SpliceAid-F[31]. Human FOXP3 target genes in Tregs were obtained[36] as were a list of proteins that interact with human FOXP3[37] and converted to AceView gene identifiers. Genes in chromosomal regions associated with risk for any one of 11 autoimmune diseases were obtained from ImmunoBase (https://genetics.opentargets.org/immunobase). For all tests, a binary response variable was used to indicate gene membership and tested using a two-sided $\chi^2$ test. Statistical significance was considered if the test attains a $P < 0.05$.

**Statistics and reproducibility**. Except where noted all statistical analyses were carried out in SAS v9.4 (SAS Institute). Equations 1, 2 and 3 were modeled using the GLIMMIX procedure. Where appropriate, statistical tests were assumed to be two-sided. All statistical tests based on symmetrically distributed test statistics were two-sided. No repeated measures data were analyzed in this study. All subjects in this study represent distinct individuals. Sample sizes used for analysis are presented in Supplementary Table 1. The Python (v3.8) packages NumPy, SciPy, Pandas, Matplotlib, and Seaborn, and the R (version 4.2) libraries RColorBrewer, scales, and ggplot2 were used in the creation of figures.

Additional annotation and assembly of figures was performed using Adobe Illustrator.

**Reporting summary**. Further information on research design is available in the Nature Portfolio Reporting Summary linked to this article.

## Data availability

The sequencing data from this study have been submitted to the NCBI Gene Expression Omnibus (GEO; https://www.ncbi.nlm.nih.gov/geo) under accession number GSE237218. All numerical source data for boxplots used in this manuscripts are available as Supplementary Data 1–9, and have been additionally deposited on FigShare (https://doi.org/10.6084/m9.figshare.22789763)[75].

## Code availability

All code pertaining to the analysis presented in this study has been deposited as a Zenodo archive (https://doi.org/10.5281/zenodo.8226066)[76], and can additionally be found at https://github.com/jrbnewman/T1D_treg_splicing/tree/master.

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

## Acknowledgements

We thank the Benaroya Research Institute (BRI) Center for Interventional Immunology, especially Kassidy Benoscek, Thien-Son Nguyen, Marli McCulloch-Olson, Jani Klein, and McKenzie Lettau, for patient recruitment and sample collection and the BRI Human Immunophenotyping Core for cell isolations and flow data analysis. Funding sources: NIDDK/1R01-DK116954 (P.C.) NIDDK/1DP3-DK085678 (P.C. and S.S.R.), R01 DK106718 (P.C.), P01AI042288 (P.C.).

## Author contributions

Conceived and designed the experiments: S.S.R., J.H.B., P.C., S.O.-G.; Samples and data were managed by: S.A.L., C.S., C.J.G., K.C., J.H.B., P.C.; Data analysis: J.R.B.N., P.C., L.M.M.; Writing - Initial draft: J.R.B.N., P.C.; Writing – review and edition: all authors.

## Competing interests

The authors declare no competing interests.
