## [Peer Review File · Communications Biology]

Reviewers' comments:

Reviewer #1 (Remarks to the Author):

The manuscript by Newman J et al. suggests changes in alternative splicing of several mRNAs of genes associated with T1D risk in Treg cells and not in memory T cells of patients with T1D. The authors relate this finding with its possible causative role in the diseases, which is an interesting finding. The study is well powered in terms of the number of patients and controls. Although I understand that this is a descriptive study, I think that the manuscript would benefit from verification of some of the differentially spliced genes also at the protein level e.g. by western blots. That would be particularly relevant for SRSF7 and TRA2B in Treg cells and show that the changes in mRNA splicing are projected into a differential expression of total proteins or their respective isoforms. My main comment is the clarity of the manuscript. The bioinformatics analyses are complicated and the authors should at least try to explain more clearly the terminology they use. This makes the manuscript very hard to read and makes the readers always bounce to the methods and back. I provide some comments to increase the manuscript's clarity. I think there should be enough space to expand the results with additional explanations. Similarly, the article would benefit from a better explanation of the methods they used in the main text to provide a better basic level of understanding to the broad audience and not only to the bioinformatic specialists. Differences between DEFU and DS are unclear as detailed in one of the comments below. In my opinion, the explanation in the text of the methods/terminology could be done similarly as was done for the multitranscript: "multi-transcript genes defined as those with at least 2 transcripts in the reduced references ". Also, the ranking method could be explained in the main text clearly and briefly. Please make sure that all the abbreviations are spelled out when first used.

Conceptually, I understand that these altered splicing may not result in remarkable differences in the overall gene expression, but I do not understand why the authors claim that they would also not result in altered function. This is difficult to understand if combined with the claim that the findings may be relevant for the disease pathogenesis.

Lastly, the studies were performed when the disease was well established (about 13 years post-onset on average): does that mean that the Treg-altered splicing is independent of the disease stage? How could that be explained from an immunological point of view? Which diagnostic implications could emerge from this study?

Specific comments:

Lane 28: "tissue-specific" There are not really different tissues analyzed in the manuscript only cell types, please consider changing it.

Lane 34 and more in the text: Etiology is the cause of a disease, but the data you provided are not considering what causes the disease. Please try to rephrase it. I understand that the motivation of the study was that the disease-associated variants are enriched in regulatory sequences, but you haven't shown any correlation/causative relationship between the mutations and the differentially spliced genes. You rather discovered a phenomenon, which may have many possible explanations (as mentioned in the discussion), thus I would rather suggest saying that your data indicate a new mechanism associated with the disease pathophysiology.

Lane 31: Why in abstract "In Tregs, 553 genes demonstrated differences in isoform usage" and there are only 402 differentially spliced genes?

Please better describe the cell isolation and please show representative FACS plots of the cell populations and purities as a supplementary figure. As you know, isolation of Treg only by CD25 expression could result in a mixed population of Treg and activated T effector cells, so it would be good to see an example of your gating and explain how you have excluded activated T cells.

Please explain what you refer to as a “reduced reference transcriptome” as e.g. defined well in the text of citation 58.

Lane 82: Please spell out APN and TPM, when it is used first in the figure legend. It makes the reader go to the methods looking for the shortcuts.

Lane 111: Please explain Ψ when first used and not only in methods.

Lane 115: What does (2) means, or is the sentence missing (1)?

Lane 119-121: Please provide conclusion for the result of the analyses “Compared to the set of DS genes in Tregs (Figure 1C), 244 of the 402 (61%) genes with differential isoform usage in Tregs also had significant differential exon usage in Tregs (Figure 1F).” The result is not explained, and it is confusing. I understand that in contrast to DS, DEFU analyses also includes genes that should not be subject to alternative splicing. But I don’t understand how can a gene have differential isoform usage (DS) and the same exon usage (DEFU)? Isn’t the difference between isoforms always caused by differential usage of exons? Or do you mean that the rest of the DS genes in T1D have retained introns and that’s what is causing them to be DS?

Lane 130-131: How figure 2A is showing this statement “In Tregs, 9% of DS genes also had significant changes in transcript rank” should it be 2B? Also please explain briefly what transcript rank means in the text before you start to use the terminology.

Lane 136: It is figure 2C not 2B.

What does figure 2A show? If the indication in the figure is not correct and 2A is 2B, 2B is 2C, where is a text explaining 2A. I didn’t understand the figure 2A at all...

Compare lanes 151 and 158: If I understood correctly this is the same type of enrichment analyses but the data presentation have a slightly different format “(12 of 95 T1D-associated region genes DS; 390 of 152 2,439 non-T1D-associated region genes DS, $P = 0.38$)” vs “(24/67 (36%) RRM-1-containing multi-transcript genes DS, 378/2467 (15%) other multi-transcript genes DS, FDR-corrected $P=0.023$)”. Please unify the format of the text.

Lane 168-169: Most of these multi-transcript splicing factor genes exhibited differential splicing between T1D cases and controls in Tregs (18/40 (45%) splicing factor genes DS, 384/2,494 (15%) of other multi-transcript genes DS, $P < 0.0001$). Correct me if I am wrong but 45% is smaller than half, and thus expression the of most of the multi-transcript splicing factor genes should be adjusted.

Figure 3, are the differences between isoform SRSF7.l, TRA2B.j and TRA2B.p expression in control versus T1D patients significant, and if yes, could you indicate that in the graph 3 and 4 respectively?

Lane 328: “~60% of expressed splicing factor genes have significant changes in isoform usage in Tregs” From the results I understood that there are 45% of splicing factors DS (Lane 168-169)

Lane 365: Please indicate the purities of the MACS isolated cells for more than one sample.

Reviewer #2 (Remarks to the Author):

In this manuscript Newman et al. performed genome wide transcript analyses to determine how transcript utilization and alternative splicing contributes to Treg cell and CD4+ non-Treg cell

compartments. The manuscript is entirely based on in silico transcriptomic characterization demonstrating isoform usage is more abundant in Tregs from T1D patients. First, as a disclaimer I should first mention that since I am not an expert of computational analyses of alternative splicing events and transcriptomics, I will not be able to comment on the robustness of the analyses provided. Taking that for granted, in my opinion the manuscript is better suited for a more specialized journal, rather than Communications Biology, which has gained much reputation in recent years for its broad readership. The manuscript is entirely in silico analyses without any functional correlative evidence on how such changes may contribute to the disease. In my opinion it is not appropriate for the journal's readership.

Reviewer #3 (Remarks to the Author):

In this manuscript, Concannon and colleagues report on their transcriptional analyses of memory CD4+ T cells from individuals with or without type 1 diabetes. The authors isolated memory CD4+ CD25- or CD4+ CD25+ T lymphocytes from patients with long-standing type 1 diabetes (mean: 14 years from diagnosis) and from control individuals. They describe that transcriptional differences overall are small, and that few genes are subject to differential expression between T1D and control samples. Notably, they identified significant differences in splicing events in a number of multi-transcript genes, specifically in CD4+CD25+ memory T cells. In particular, the authors note that differentially spliced transcripts include many with RNA recognition motifs and a majority of splicing factors. The results suggest a change in the splicing machinery that could underlie broader change in overall splicing within CD4+CD25+ cells.

The authors go on to describe in more detail two examples of multi-transcript genes (SRSF7 and TRAB2) to illustrate the type of differential splicing observed in their study.

Overall, the manuscript is well written, presents the data very clearly, and the authors' interpretations are balanced and supported by the data presented. The illustrations are clear and include useful schematics to explain both the approach and the data. I have no concerns with either the data analyses or the presentation that are rigorous and logical.

To improve the manuscript, I would suggest including Supplementary Fig S3 into the main manuscript or at least referencing to this figure at the very beginning of the results section. This helpful schematic is only referenced in the methods section, where it will likely go unnoticed by most readers. I would further suggest that the authors include a brief description of patient characteristics (summarized in Supplementary table 2) at the beginning of the manuscript where no mention is made of these data. In addition, it would be pertinent to provide the rationale for focusing on memory CD4 T cells among all the possible immunocyte populations that could be relevant to T1D. There is a clear argument to select a small number of cell types for comparison and to prioritize memory CD4 T cells, but this argument should be made at the outset. Otherwise, the choice of these particular subsets could seem arbitrary.

I find the discussion informative and balanced, and the overall findings of interest as they highlight an aspect of transcriptional differences relevant to T1D, and autoimmune diseases in general, that have not previously been given much attention. As such, this manuscript should be of interest to the field.

Newman et al. response to critiques (responses in italics)

Reviewers' comments:

Reviewer #1 (Remarks to the Author):

The manuscript by Newman J et al. suggests changes in alternative splicing of several mRNAs of genes associated with T1D risk in Treg cells and not in memory T cells of patients with T1D. The authors relate this finding with its possible causative role in the diseases, which is an interesting finding. The study is well powered in terms of the number of patients and controls. Although I understand that this is a descriptive study, I think that the manuscript would benefit from verification of some of the differentially spliced genes also at the protein level e.g. by western blots. That would be particularly relevant for SRSF7 and TRA2B in Treg cells and show that the changes in mRNA splicing are projected into a differential expression of total proteins or their respective isoforms.

We appreciate the reviewer's concern, which is also raised by Reviewer #2. Tregs are a minor cell population in peripheral blood, constituting 1% to 4% of nucleated cells. Thus cells are limiting for our analyses and we were unable to prepare contemporaneous protein lysates that would be necessary for the requested immunoblotting. In addition, we were unable to identify any commercially available antibodies that detect the products of poison-cassette containing transcripts, likely due to their turnover by nonsense-mediated decay. Functional studies of poison-cassette exon usage typically utilize tagged constructs in order to address this issue. Splicing factor genes including SRSF1, SRSF2, SRSF3, SRSF5, SRSF7 and TRA2B have been documented in published studies to display both altered transcript and protein levels in multiple cell types driven by differential poison cassette exon incorporation. We now cite these studies in the manuscript.

As an alternative approach to provide validation for the effects of the observed differences in gene expression at the protein level, we turned to flow cytometry. We measured the expression of several immune markers in Tregs sorted from the same PBMC samples derived from the same subjects. While a direct MFI-to-RNA level correlation may not be appropriate given that the cells used to collect immunophenotyping data were purified by flow cytometry (CD4⁺/CD25⁺/CD127^{lo}) as opposed to the bead-purification used to isolate cells for RNA extraction, we did compare the effect sizes and directions between specific markers and their corresponding genes. We found agreement between the two sets of data in Tregs, suggesting that, at least in this limited sampling, the differences observed at the RNA level reflect the differences in the corresponding proteins. In particular we quantified FOXP3 expression at both the RNA and protein level and now report these data in the text. Additional marker data are provided in the Supplementary Data section.

The new FOXP3 data provided an additional insight not noted in the original submission. FOXP3 RNA and protein (measured by MFI) were modestly, but significantly, elevated in T1D cases as compared to controls. Nine of the DS splicing factor genes noted in our study, including four where their designation as DS is due to differential incorporation of a poison cassette exon, are direct targets of FOXP3 transcriptional regulation. Thus, some portion of the splicing dysregulation we observe in Tregs from T1D cases may reflect the impact of this increased FOXP3 expression driving splicing factor gene expression leading to PCE incorporation. We now comment on this in the discussion where we consider different explanations for our observations.

My main comment is the clarity of the manuscript. The bioinformatics analyses are complicated and the authors should at least try to explain more clearly the terminology they use. This makes the manuscript very hard to read and makes the readers always bounce to the methods and back. I provide some comments to increase the manuscript's clarity. I think there should be enough space to expand the results with additional explanations. Similarly, the article would benefit from a better explanation of the methods they used in the main text to provide a better basic level of understanding to the broad audience and not only to the bioinformatic specialists.

We have reworked and expanded the text as the reviewer suggests below.

Differences between DEFU and DS are unclear as detailed in one of the comments below. In my opinion, the explanation in the text of the methods/terminology could be done similarly as was done for the multitranscript: “multi-transcript genes defined as those with at least 2 transcripts in the reduced references “.

DS tests whether the distribution of transcript levels within a gene varies between cases and controls and DEFU tests the distribution of the levels of individual exonic sequences within a gene varies between cases and controls. We have added text to the results to help differentiate the two analyses:

“We next examined if there was evidence of differential exon fragment usage (DEFU) between TID cases and controls. This is a variation on the test for differential splicing: while the gene-based DS test is based on the distribution of transcript levels within a gene and how this varies between TID cases and controls, the test for DEFU considers if the distribution of exon levels within a gene vary.”

Also, the ranking method could be explained in the main text clearly and briefly. Please make sure that all the abbreviations are spelled out when first used.

Done. The beginning of the Results subsection “Differential splicing alters isoform usage” now reads:

“In addition to determining whether isoforms are expressed at different levels, we also assessed whether the most common isoform changed between cases and controls by binning the transcripts of each gene into three ranks: rank 1 containing the most expressed transcript(s) of that gene, rank 3 comprising the least expressed transcripts(s), and rank 2 consisting of all other expressed transcripts.”

Conceptually, I understand that these altered splicing may not result in remarkable differences in the overall gene expression, but I do not understand why the authors claim that they would also not result in altered function. This is difficult to understand if combined with the claim that the findings may be relevant for the disease pathogenesis.

While we have tried to be cautious in our interpretation of the functional consequences of altered splicing patterns we see in comparing Tregs from TID cases and controls, it was not our intent to argue against possible effects on protein function. We offer several possible effects of

alternative splicing in Tregs in the discussion, one of which invokes loss of function as a mechanism:

“Shifts in isoform usage could serve to regulate gene expression at a post-transcriptional level by diverting proportions of transcripts for a given gene to isoforms that fail to produce fully functional protein.”

We have previously documented this mechanism in detailed studies of a T1D risk variant in the SIRPG gene that disrupts an exonic splicing enhancer. However, in the discussion we also cite the possibility of altered or diminished function as a possible outcome of alternative splicing:

“Alternatively, transcripts could produce truncated or altered proteins with the potential to disrupt cellular processes by dominant interference.”

Lastly, the studies were performed when the disease was well established (about 13 years post-onset on average): does that mean that the Treg-altered splicing is independent of the disease stage? How could that be explained from an immunological point of view? Which diagnostic implications could emerge from this study?

The reviewer raises a good point that we had not commented on in the paper. We now acknowledge that the effects we observe are not limited to the active phase of the disease during which islet destruction is ongoing. This is not unusual among effects controlled by T1D risk loci, many of which can be associated with allelic differences in gene expression, or protein function, both in T1D patients long after diagnosis or in otherwise healthy controls. In some cases, such as the rs6043409 SNP in SIRPG, the effects are mediated via splicing. Ideally, a study of those at risk followed through onset (such as the TEDDY Study) would be required to fully address this issue where longitudinal sampling could be coupled with differences in gene expression/protein function

Specific comments:

Lane 28: “tissue-specific” There are not really different tissues analyzed in the manuscript only cell types, please consider changing it.

As suggested, we have corrected this, and for clarity, have expanded it to read “cell-type-specific representative transcriptomes”

Lane 34 and more in the text: Etiology is the cause of a disease, but the data you provided are not considering what causes the disease. Please try to rephrase it. I understand that the motivation of the study was that the disease-associated variants are enriched in regulatory sequences, but you haven't shown any correlation/causative relationship between the mutations and the differentially spliced genes. You rather discovered a phenomenon, which may have many possible explanations (as mentioned in the discussion), thus I would rather suggest saying that your data indicate a new mechanism associated with the disease pathophysiology.

We have edited this to now read “Our results suggest that dysregulation of gene expression, through shifts in alternative splicing in Tregs, as a novel phenomenon associated with T1D pathophysiology.”

Lane 31: Why in abstract “In Tregs, 553 genes demonstrated differences in isoform usage” and there are only 402 differentially spliced genes?

We apologize for this oversight. The incorrect numbers, cited in the abstract, were not caught during proofing. The numbers cited in the results section are the correct counts and are consistent throughout the manuscript.

Please better describe the cell isolation and please show representative FACS plots of the cell populations and purities as a supplementary figure. As you know, isolation of Treg only by CD25 expression could result in a mixed population of Treg and activated T effector cells, so it would be good to see an example of your gating and explain how you have excluded activated T cells.

A supplementary figure showing the gating as well as a plot of cell purities measured from multiple samples taken at different time points during the project is now provided as Supplementary Figure S4.

Please explain what you refer to as a “reduced reference transcriptome” as e.g. defined well in the text of citation 58.

A “reduced reference transcriptome” refers to an approximation of the set of expressed transcripts (based on an annotated database of known and predicted transcripts) for which there is evidence of its complete expression. This is a data-driven approach that segments genes into their constitutive exons and exon-exon junctions; a transcript is excluded on the basis that there are one or more of their exons or junctions without sufficient sequencing coverage supporting their transcription.

We have added this text to the beginning of the results section, with an appropriate reference.

Lane 82: Please spell out APN and TPM, when it is used first in the figure legend. It makes the reader go to the methods looking for the shortcuts.

This has been done.

Lane 111: Please explain Ψ when first used and not only in methods.

This has been done.

Lane 115: What does (2) means, or is the sentence missing (1)?

The sentence was missing (1). This has now been corrected.

Lane 119-121: Please provide conclusion for the result of the analyses “Compared

to the set of DS genes in Tregs (Figure 1C), 244 of the 402 (61%) genes with differential isoform usage in Tregs also had significant differential exon usage in Tregs (Figure 1F).” The result is not explained, and it is confusing. I understand that in contrast to DS, DEFU analyses also includes genes that should not be subject to alternative splicing. But I don’t understand how can a gene have differential isoform usage (DS) and the same exon usage (DEFU)? Isn’t the difference between isoforms always caused by differential usage of exons? Or do you mean that the rest of the DS genes in T1D have retained introns and that’s what is causing them to be DS?

Done. We have added a brief explanation of the difference between DS and DEFU tests: “This is a variation on the test for differential splicing: while the gene-based DS test is based on the distribution of transcript levels within a gene and how this varies between T1D cases and controls, the test for DEFU considers if the distribution of exon levels within a gene vary.” as well as a short concluding statement: “suggesting that changes in isoform structure are abundant in Tregs in T1D”

There are several possible explanations for why a gene in our study might have differential isoform usage (DS) yet still not display significant DEFU. The most likely is that it reflects the measure of statistical significance as a gene that is significantly DS may not be DEFU (and vice versa) if it does not pass this threshold. Alternatively, there may be exons of a gene that are expressed but are not included in the reduced reference transcriptomes because their parent transcripts do not have sufficient evidence of expression. The DEFU test would not exclude these exons. There might also may be the case where exon levels do not substantially change but there is a change in isoform abundance estimates.

Lane 130-131: How figure 2A is showing this statement “In Tregs, 9% of DS genes also had significant changes in transcript rank” should it be 2B? Also please explain briefly what transcript rank means in the text before you start to use the terminology.

This should have been figure 2B, and has now been corrected. We also now explain Figure 2A and the transcript rank terminology at the beginning of this section, which details hypothetical scenarios where a gene can be considered differentially spliced (quantitative changes in transcript abundance) vs changes in transcript rank (how frequently a transcript is among the most expressed/least expressed/other). We have added this text:

“Figure 3A shows a hypothetical example of the difference between the DS test and changes in transcript rank. A gene may have a statistically significant quantitative difference in transcript levels between T1D cases and controls, but transcripts may not necessarily deviate substantially in their transcript rank (Figure 3A, first panel). Alternatively, there may a scenario where the distribution of transcript levels of a gene do not significantly differ between conditions (e.g. similar mean expression and/or high variance) and therefore is not DS, but there may be a significant change in how frequently transcripts are ranked (Figure 3A, second panel). A gene may be considered DS and also have differentially-ranked transcripts if there is a significant quantitative difference in transcript abundances and also a shift in transcript rank frequency (Figure 3A, third panel).”

Lane 136: It is figure 2C not 2B.

Correction has been made.

What does figure 2A show? If the indication in the figure is not correct and 2A is 2B, 2B is 2C, where is a text explaining 2A. I didn't understand the figure 2A at all...

Figure 2A (now Figure 3A) is an explanation of the differences between the test differential splicing and changes in transcript rank. As stated above we had added the following text to further explain this figure:

“Figure 3A shows a hypothetical example of the difference between the DS test and changes in transcript rank. A gene may have a statistically significant quantitative difference in transcript levels between T1D cases and controls, but transcripts may not necessarily deviate substantially in their transcript rank (Figure 3A, first panel). Alternatively, there may a scenario where the distribution of transcript levels of a gene do not significant differ between conditions (e.g. similar mean expression and/or high variance) and therefore is not DS, but there may be a significant change in the how frequently transcripts are ranked (Figure 3A, second panel). A gene may be considered DS and also have differentially-ranked transcripts if there is a significant quantitative difference in transcript abundances and also a shift in transcript rank frequency (Figure 3A, third panel).”

Compare lanes 151 and 158: If I understood correctly this is the same type of enrichment analyses but the data presentation have a slightly different format “(12 of 95 T1D-associated region genes DS; 390 of 152 2,439 non-T1D-associated region genes DS, $P = 0.38$)” vs “(24/67 (36%) RRM-1-containing multi-transcript genes DS, 378/2467 (15%) other multi-transcript genes DS, FDR-corrected $P=0.023$)”. Please unify the format of the text.

This has been corrected so that now all text pertaining to enrichments and proportions is formatted identically.

Lane 168-169: Most of these multi-transcript splicing factor genes exhibited differential splicing between T1D cases and controls in Tregs (18/40 (45%) splicing factor genes DS, 384/2,494 (15%) of other multi-transcript genes DS, $P < 0.0001$). Correct me if I am wrong but 45% is smaller than half, and thus expression the of most of the multi-transcript splicing factor genes should be adjusted.

This has been corrected.

Figure 3, are the differences between isoform SRSF7.1, TRA2B.j and TRA2B.p expression in control versus T1D patients significant, and if yes, could you indicate that in the graph 3 and 4 respectively?

For SRSF7.1 and TRA2B.p the difference in transcript levels between cases and controls is nominally significant using a similar model for testing gene differential expression. This has been indicated in their appropriate figures. We also provide statistics on the percent-spliced-in difference for SRSF7 and TRA2B in panel E of their respective figure.

Lane 328: “~60% of expressed splicing factor genes have significant changes in isoform usage in Tregs” From the results I understood that there are 45% of splicing factors DS (Lane 168-169)

This has been corrected.

Lane 365: Please indicate the purities of the MACS isolated cells for more than one sample.

We now provide in the supplementary data (Supplementary Figure S4) a plot of purities of the memory CD4⁺CD25⁺ cells sampled at different times during the project with separate results for T1D cases and controls.

Reviewer #2 (Remarks to the Author):

In this manuscript Newman et al. performed genome wide transcript analyses to determine how transcript utilization and alternative splicing contributes to Treg cell and CD4⁺ non-Treg cell compartments. The manuscript is entirely based on in silico transcriptomic characterization demonstrating isoform usage is more abundant in Tregs from T1D patients. First, as a disclaimer I should first mention that since I am not an expert of computational analyses of alternative splicing events and transcriptomics, I will not be able to comment on the robustness of the analyses provided. Taking that for granted, in my opinion the manuscript is better suited for a more specialized journal, rather than Communications Biology, which has gained much reputation in recent years for its broad readership. The manuscript is entirely in silico analyses without any functional correlative evidence on how such changes may contribute to the disease. In my opinion it is not appropriate for the journal's readership.

We appreciate the time the reviewer has taken to consider the work. We do agree that functional (protein) evidence would be useful addition to our findings. However, as we note in our response to Reviewer #1, many of the changes in alternative splicing seemingly relate to the increased production of RNA transcripts that do not likely code for proteins (e.g. intron retention, poison cassette exon inclusion), and that the protein-coding forms do not appear to be greatly affected. That being said, in this revised version of the manuscript, we now include flow cytometry measurements of several proteins, notably FOXP3, demonstrating that the direction of differences and the magnitudes are similar when comparing cases and controls by RNA-seq or FACS.

Reviewer #3 (Remarks to the Author):

In this manuscript, Concannon and colleagues report on their transcriptional analyses of memory CD4⁺ T cells from individuals with or without type 1 diabetes. The authors isolated memory CD4⁺ CD25⁻ or CD4⁺ CD25⁺ T lymphocytes from patients with long-standing type 1 diabetes (mean: 14 years from diagnosis) and from control individuals. They describe that transcriptional differences overall are small, and that few genes are subject to differential expression between T1D and control samples. Notably, they identified significant differences in splicing events in a number of multi-transcript genes, specifically in CD4⁺CD25⁺ memory T cells. In particular, the

authors note that differentially spliced transcripts include many with RNA recognition motifs and a majority of splicing factors. The results suggest a change in the splicing machinery that could underlie broader change in overall splicing within CD4+CD25+ cells.

The authors go on to describe in more detail two examples of multi-transcript genes (SRSF7 and TRAB2) to illustrate the type of differential splicing observed in their study.

Overall, the manuscript is well written, presents the data very clearly, and the authors' interpretations are balanced and supported by the data presented. The illustrations are clear and include useful schematics to explain both the approach and the data. I have no concerns with either the data analyses or the presentation that are rigorous and logical.

To improve the manuscript, I would suggest including Supplementary Fig S3 into the main manuscript or at least referencing to this figure at the very beginning of the results section. This helpful schematic is only referenced in the methods section, where it will likely go unnoticed by most readers. I would further suggest that the authors include a brief description of patient characteristics (summarized in Supplementary table 2) at the beginning of the manuscript where no mention is made of these data.

We agree with these suggestions and have made the requested changes. Supplementary Fig S3 is now Figure 1, and subsequent figures have been renumbered. A short description of patient characteristics is now also included at the beginning of the Results section.

In addition, it would be pertinent to provide the rationale for focusing on memory CD4 T cells among all the possible immunocyte populations that could be relevant to T1D. There is a clear argument to select a small number of cell types for comparison and to prioritize memory CD4 T cells, but this argument should be made at the outset. Otherwise, the choice of these particular subsets could seem arbitrary.

While we had previously only commented on the choice of cell types in the discussion, this comment, as well as similar concerns raised by Reviewer #1 clearly indicate that a greater emphasis on this issue is necessary. We have added text to the introduction of the paper indicating that autoreactive memory T cells can both contribute to disease progression and potentially limit attempts to transplant islets as a treatment. We also highlight the role of Tregs in controlling autoimmunity.

I find the discussion informative and balanced, and the overall findings of interest as they highlight an aspect of transcriptional differences relevant to T1D, and autoimmune diseases in general, that have not previously been given much attention. As such, this manuscript should be of interest to the field.

We thank the reviewer for their comments and appreciation of the work.

REVIEWERS' COMMENTS:

Reviewer #1 (Remarks to the Author):

The Authors significantly worked on improving the manuscript, which has gained clarity and functional insights.

Reviewer #3 (Remarks to the Author):

The authors have responded to my suggestions with edits that have further improved this excellent manuscript and I have no further comments.

Newman et al. response to critiques (responses in italics)

Reviewers' comments:

Reviewer #1 (Remarks to the Author):

The manuscript by Newman J et al. suggests changes in alternative splicing of several mRNAs of genes associated with T1D risk in Treg cells and not in memory T cells of patients with T1D. The authors relate this finding with its possible causative role in the diseases, which is an interesting finding. The study is well powered in terms of the number of patients and controls. Although I understand that this is a descriptive study, I think that the manuscript would benefit from verification of some of the differentially spliced genes also at the protein level e.g. by western blots. That would be particularly relevant for SRSF7 and TRA2B in Treg cells and show that the changes in mRNA splicing are projected into a differential expression of total proteins or their respective isoforms.

We appreciate the reviewer's concern, which is also raised by Reviewer #2. Tregs are a minor cell population in peripheral blood, constituting 1% to 4% of nucleated cells. Thus cells are limiting for our analyses and we were unable to prepare contemporaneous protein lysates that would be necessary for the requested immunoblotting. In addition, we were unable to identify any commercially available antibodies that detect the products of poison-cassette containing transcripts, likely due to their turnover by nonsense-mediated decay. Functional studies of poison-cassette exon usage typically utilize tagged constructs in order to address this issue. Splicing factor genes including SRSF1, SRSF2, SRSF3, SRSF5, SRSF7 and TRA2B have been documented in published studies to display both altered transcript and protein levels in multiple cell types driven by differential poison cassette exon incorporation. We now cite these studies in the manuscript.

As an alternative approach to provide validation for the effects of the observed differences in gene expression at the protein level, we turned to flow cytometry. We measured the expression of several immune markers in Tregs sorted from the same PBMC samples derived from the same subjects. While a direct MFI-to-RNA level correlation may not be appropriate given that the cells used to collect immunophenotyping data were purified by flow cytometry ($CD4^+/CD25^+/CD127^lo$) as opposed to the bead-purification used to isolate cells for RNA extraction, we did compare the effect sizes and directions between specific markers and their corresponding genes. We found agreement between the two sets of data in Tregs, suggesting that, at least in this limited sampling, the differences observed at the RNA level reflect the differences in the corresponding proteins. In particular we quantified FOXP3 expression at both the RNA and protein level and now report these data in the text. Additional marker data are provided in the Supplementary Data section.

The new FOXP3 data provided an additional insight not noted in the original submission. FOXP3 RNA and protein (measured by MFI) were modestly, but significantly, elevated in T1D cases as compared to controls. Nine of the DS splicing factor genes noted in our study, including four where their designation as DS is due to differential incorporation of a poison cassette exon, are direct targets of FOXP3 transcriptional regulation. Thus, some portion of the splicing dysregulation we observe in Tregs from T1D cases may reflect the impact of this increased FOXP3 expression driving splicing factor gene expression leading to PCE incorporation. We now comment on this in the discussion where we consider different explanations for our observations.

My main comment is the clarity of the manuscript. The bioinformatics analyses are complicated and the authors should at least try to explain more clearly the terminology they use. This makes the manuscript very hard to read and makes the readers always bounce to the methods and back. I provide some comments to increase the manuscript's clarity. I think there should be enough space to expand the results with additional explanations. Similarly, the article would benefit from a better explanation of the methods they used in the main text to provide a better basic level of understanding to the broad audience and not only to the bioinformatic specialists.

We have reworked and expanded the text as the reviewer suggests below.

Differences between DEFU and DS are unclear as detailed in one of the comments below. In my opinion, the explanation in the text of the methods/terminology could be done similarly as was done for the multitranscript: “multi-transcript genes defined as those with at least 2 transcripts in the reduced references “.

DS tests whether the distribution of transcript levels within a gene varies between cases and controls and DEFU tests the distribution of the levels of individual exonic sequences within a gene varies between cases and controls. We have added text to the results to help differentiate the two analyses:

“We next examined if there was evidence of differential exon fragment usage (DEFU) between TID cases and controls. This is a variation on the test for differential splicing: while the gene-based DS test is based on the distribution of transcript levels within a gene and how this varies between TID cases and controls, the test for DEFU considers if the distribution of exon levels within a gene vary.”

Also, the ranking method could be explained in the main text clearly and briefly. Please make sure that all the abbreviations are spelled out when first used.

Done. The beginning of the Results subsection “Differential splicing alters isoform usage” now reads:

“In addition to determining whether isoforms are expressed at different levels, we also assessed whether the most common isoform changed between cases and controls by binning the transcripts of each gene into three ranks: rank 1 containing the most expressed transcript(s) of that gene, rank 3 comprising the least expressed transcripts(s), and rank 2 consisting of all other expressed transcripts.”

Conceptually, I understand that these altered splicing may not result in remarkable differences in the overall gene expression, but I do not understand why the authors claim that they would also not result in altered function. This is difficult to understand if combined with the claim that the findings may be relevant for the disease pathogenesis.

While we have tried to be cautious in our interpretation of the functional consequences of altered splicing patterns we see in comparing Tregs from TID cases and controls, it was not our intent to argue against possible effects on protein function. We offer several possible effects of

alternative splicing in Tregs in the discussion, one of which invokes loss of function as a mechanism:

“Shifts in isoform usage could serve to regulate gene expression at a post-transcriptional level by diverting proportions of transcripts for a given gene to isoforms that fail to produce fully functional protein.”

We have previously documented this mechanism in detailed studies of a T1D risk variant in the SIRPG gene that disrupts an exonic splicing enhancer. However, in the discussion we also cite the possibility of altered or diminished function as a possible outcome of alternative splicing:

“Alternatively, transcripts could produce truncated or altered proteins with the potential to disrupt cellular processes by dominant interference.”

Lastly, the studies were performed when the disease was well established (about 13 years post-onset on average): does that mean that the Treg-altered splicing is independent of the disease stage? How could that be explained from an immunological point of view? Which diagnostic implications could emerge from this study?

The reviewer raises a good point that we had not commented on in the paper. We now acknowledge that the effects we observe are not limited to the active phase of the disease during which islet destruction is ongoing. This is not unusual among effects controlled by T1D risk loci, many of which can be associated with allelic differences in gene expression, or protein function, both in T1D patients long after diagnosis or in otherwise healthy controls. In some cases, such as the rs6043409 SNP in SIRPG, the effects are mediated via splicing. Ideally, a study of those at risk followed through onset (such as the TEDDY Study) would be required to fully address this issue where longitudinal sampling could be coupled with differences in gene expression/protein function

Specific comments:

Lane 28: “tissue-specific” There are not really different tissues analyzed in the manuscript only cell types, please consider changing it.

As suggested, we have corrected this, and for clarity, have expanded it to read “cell-type-specific representative transcriptomes”

Lane 34 and more in the text: Etiology is the cause of a disease, but the data you provided are not considering what causes the disease. Please try to rephrase it. I understand that the motivation of the study was that the disease-associated variants are enriched in regulatory sequences, but you haven't shown any correlation/causative relationship between the mutations and the differentially spliced genes. You rather discovered a phenomenon, which may have many possible explanations (as mentioned in the discussion), thus I would rather suggest saying that your data indicate a new mechanism associated with the disease pathophysiology.

We have edited this to now read “Our results suggest that dysregulation of gene expression, through shifts in alternative splicing in Tregs, as a novel phenomenon associated with T1D pathophysiology.”

Lane 31: Why in abstract “In Tregs, 553 genes demonstrated differences in isoform usage” and there are only 402 differentially spliced genes?

We apologize for this oversight. The incorrect numbers, cited in the abstract, were not caught during proofing. The numbers cited in the results section are the correct counts and are consistent throughout the manuscript.

Please better describe the cell isolation and please show representative FACS plots of the cell populations and purities as a supplementary figure. As you know, isolation of Treg only by CD25 expression could result in a mixed population of Treg and activated T effector cells, so it would be good to see an example of your gating and explain how you have excluded activated T cells.

A supplementary figure showing the gating as well as a plot of cell purities measured from multiple samples taken at different time points during the project is now provided as Supplementary Figure S4.

Please explain what you refer to as a “reduced reference transcriptome” as e.g. defined well in the text of citation 58.

A “reduced reference transcriptome” refers to an approximation of the set of expressed transcripts (based on an annotated database of known and predicted transcripts) for which there is evidence of its complete expression. This is a data-driven approach that segments genes into their constitutive exons and exon-exon junctions; a transcript is excluded on the basis that there are one or more of their exons or junctions without sufficient sequencing coverage supporting their transcription.

We have added this text to the beginning of the results section, with an appropriate reference.

Lane 82: Please spell out APN and TPM, when it is used first in the figure legend. It makes the reader go to the methods looking for the shortcuts.

This has been done.

Lane 111: Please explain Ψ when first used and not only in methods.

This has been done.

Lane 115: What does (2) means, or is the sentence missing (1)?

The sentence was missing (1). This has now been corrected.

Lane 119-121: Please provide conclusion for the result of the analyses “Compared

to the set of DS genes in Tregs (Figure 1C), 244 of the 402 (61%) genes with differential isoform usage in Tregs also had significant differential exon usage in Tregs (Figure 1F).” The result is not explained, and it is confusing. I understand that in contrast to DS, DEFU analyses also includes genes that should not be subject to alternative splicing. But I don’t understand how can a gene have differential isoform usage (DS) and the same exon usage (DEFU)? Isn’t the difference between isoforms always caused by differential usage of exons? Or do you mean that the rest of the DS genes in T1D have retained introns and that’s what is causing them to be DS?

Done. We have added a brief explanation of the difference between DS and DEFU tests: “This is a variation on the test for differential splicing: while the gene-based DS test is based on the distribution of transcript levels within a gene and how this varies between T1D cases and controls, the test for DEFU considers if the distribution of exon levels within a gene vary.” as well as a short concluding statement: “suggesting that changes in isoform structure are abundant in Tregs in T1D”

There are several possible explanations for why a gene in our study might have differential isoform usage (DS) yet still not display significant DEFU. The most likely is that it reflects the measure of statistical significance as a gene that is significantly DS may not be DEFU (and vice versa) if it does not pass this threshold. Alternatively, there may be exons of a gene that are expressed but are not included in the reduced reference transcriptomes because their parent transcripts do not have sufficient evidence of expression. The DEFU test would not exclude these exons. There might also may be the case where exon levels do not substantially change but there is a change in isoform abundance estimates.

Lane 130-131: How figure 2A is showing this statement “In Tregs, 9% of DS genes also had significant changes in transcript rank” should it be 2B? Also please explain briefly what transcript rank means in the text before you start to use the terminology.

This should have been figure 2B, and has now been corrected. We also now explain Figure 2A and the transcript rank terminology at the beginning of this section, which details hypothetical scenarios where a gene can be considered differentially spliced (quantitative changes in transcript abundance) vs changes in transcript rank (how frequently a transcript is among the most expressed/least expressed/other). We have added this text:

“Figure 3A shows a hypothetical example of the difference between the DS test and changes in transcript rank. A gene may have a statistically significant quantitative difference in transcript levels between T1D cases and controls, but transcripts may not necessarily deviate substantially in their transcript rank (Figure 3A, first panel). Alternatively, there may a scenario where the distribution of transcript levels of a gene do not significantly differ between conditions (e.g. similar mean expression and/or high variance) and therefore is not DS, but there may be a significant change in how frequently transcripts are ranked (Figure 3A, second panel). A gene may be considered DS and also have differentially-ranked transcripts if there is a significant quantitative difference in transcript abundances and also a shift in transcript rank frequency (Figure 3A, third panel).”

Lane 136: It is figure 2C not 2B.

Correction has been made.

What does figure 2A show? If the indication in the figure is not correct and 2A is 2B, 2B is 2C, where is a text explaining 2A. I didn't understand the figure 2A at all...

Figure 2A (now Figure 3A) is an explanation of the differences between the test differential splicing and changes in transcript rank. As stated above we had added the following text to further explain this figure:

“Figure 3A shows a hypothetical example of the difference between the DS test and changes in transcript rank. A gene may have a statistically significant quantitative difference in transcript levels between T1D cases and controls, but transcripts may not necessarily deviate substantially in their transcript rank (Figure 3A, first panel). Alternatively, there may a scenario where the distribution of transcript levels of a gene do not significant differ between conditions (e.g. similar mean expression and/or high variance) and therefore is not DS, but there may be a significant change in the how frequently transcripts are ranked (Figure 3A, second panel). A gene may be considered DS and also have differentially-ranked transcripts if there is a significant quantitative difference in transcript abundances and also a shift in transcript rank frequency (Figure 3A, third panel).”

Compare lanes 151 and 158: If I understood correctly this is the same type of enrichment analyses but the data presentation have a slightly different format “(12 of 95 T1D-associated region genes DS; 390 of 152 2,439 non-T1D-associated region genes DS, $P = 0.38$)” vs “(24/67 (36%) RRM-1-containing multi-transcript genes DS, 378/2467 (15%) other multi-transcript genes DS, FDR-corrected $P=0.023$)”. Please unify the format of the text.

This has been corrected so that now all text pertaining to enrichments and proportions is formatted identically.

Lane 168-169: Most of these multi-transcript splicing factor genes exhibited differential splicing between T1D cases and controls in Tregs (18/40 (45%) splicing factor genes DS, 384/2,494 (15%) of other multi-transcript genes DS, $P < 0.0001$). Correct me if I am wrong but 45% is smaller than half, and thus expression the of most of the multi-transcript splicing factor genes should be adjusted.

This has been corrected.

Figure 3, are the differences between isoform SRSF7.1, TRA2B.j and TRA2B.p expression in control versus T1D patients significant, and if yes, could you indicate that in the graph 3 and 4 respectively?

For SRSF7.1 and TRA2B.p the difference in transcript levels between cases and controls is nominally significant using a similar model for testing gene differential expression. This has been indicated in their appropriate figures. We also provide statistics on the percent-spliced-in difference for SRSF7 and TRA2B in panel E of their respective figure.

Lane 328: “~60% of expressed splicing factor genes have significant changes in isoform usage in Tregs” From the results I understood that there are 45% of splicing factors DS (Lane 168-169)

This has been corrected.

Lane 365: Please indicate the purities of the MACS isolated cells for more than one sample.

We now provide in the supplementary data (Supplementary Figure S4) a plot of purities of the memory CD4⁺CD25⁺ cells sampled at different times during the project with separate results for T1D cases and controls.

Reviewer #2 (Remarks to the Author):

In this manuscript Newman et al. performed genome wide transcript analyses to determine how transcript utilization and alternative splicing contributes to Treg cell and CD4⁺ non-Treg cell compartments. The manuscript is entirely based on in silico transcriptomic characterization demonstrating isoform usage is more abundant in Tregs from T1D patients. First, as a disclaimer I should first mention that since I am not an expert of computational analyses of alternative splicing events and transcriptomics, I will not be able to comment on the robustness of the analyses provided. Taking that for granted, in my opinion the manuscript is better suited for a more specialized journal, rather than Communications Biology, which has gained much reputation in recent years for its broad readership. The manuscript is entirely in silico analyses without any functional correlative evidence on how such changes may contribute to the disease. In my opinion it is not appropriate for the journal's readership.

We appreciate the time the reviewer has taken to consider the work. We do agree that functional (protein) evidence would be useful addition to our findings. However, as we note in our response to Reviewer #1, many of the changes in alternative splicing seemingly relate to the increased production of RNA transcripts that do not likely code for proteins (e.g. intron retention, poison cassette exon inclusion), and that the protein-coding forms do not appear to be greatly affected. That being said, in this revised version of the manuscript, we now include flow cytometry measurements of several proteins, notably FOXP3, demonstrating that the direction of differences and the magnitudes are similar when comparing cases and controls by RNA-seq or FACS.

Reviewer #3 (Remarks to the Author):

In this manuscript, Concannon and colleagues report on their transcriptional analyses of memory CD4⁺ T cells from individuals with or without type 1 diabetes. The authors isolated memory CD4⁺ CD25⁻ or CD4⁺ CD25⁺ T lymphocytes from patients with long-standing type 1 diabetes (mean: 14 years from diagnosis) and from control individuals. They describe that transcriptional differences overall are small, and that few genes are subject to differential expression between T1D and control samples. Notably, they identified significant differences in splicing events in a number of multi-transcript genes, specifically in CD4⁺CD25⁺ memory T cells. In particular, the

authors note that differentially spliced transcripts include many with RNA recognition motifs and a majority of splicing factors. The results suggest a change in the splicing machinery that could underlie broader change in overall splicing within CD4+CD25+ cells.

The authors go on to describe in more detail two examples of multi-transcript genes (SRSF7 and TRAB2) to illustrate the type of differential splicing observed in their study.

Overall, the manuscript is well written, presents the data very clearly, and the authors' interpretations are balanced and supported by the data presented. The illustrations are clear and include useful schematics to explain both the approach and the data. I have no concerns with either the data analyses or the presentation that are rigorous and logical.

To improve the manuscript, I would suggest including Supplementary Fig S3 into the main manuscript or at least referencing to this figure at the very beginning of the results section. This helpful schematic is only referenced in the methods section, where it will likely go unnoticed by most readers. I would further suggest that the authors include a brief description of patient characteristics (summarized in Supplementary table 2) at the beginning of the manuscript where no mention is made of these data.

We agree with these suggestions and have made the requested changes. Supplementary Fig S3 is now Figure 1, and subsequent figures have been renumbered. A short description of patient characteristics is now also included at the beginning of the Results section.

In addition, it would be pertinent to provide the rationale for focusing on memory CD4 T cells among all the possible immunocyte populations that could be relevant to T1D. There is a clear argument to select a small number of cell types for comparison and to prioritize memory CD4 T cells, but this argument should be made at the outset. Otherwise, the choice of these particular subsets could seem arbitrary.

While we had previously only commented on the choice of cell types in the discussion, this comment, as well as similar concerns raised by Reviewer #1 clearly indicate that a greater emphasis on this issue is necessary. We have added text to the introduction of the paper indicating that autoreactive memory T cells can both contribute to disease progression and potentially limit attempts to transplant islets as a treatment. We also highlight the role of Tregs in controlling autoimmunity.

I find the discussion informative and balanced, and the overall findings of interest as they highlight an aspect of transcriptional differences relevant to T1D, and autoimmune diseases in general, that have not previously been given much attention. As such, this manuscript should be of interest to the field.

We thank the reviewer for their comments and appreciation of the work.